# An increase in neural stem cells and olfactory bulb adult neurogenesis improves discrimination of highly similar odorants

Sara Bragado Alonso[1], Janine K Reinert[2], Nicolas Marichal[3,4], Simone Massalini[1], Benedikt Berninger[3,4] (iD), Thomas Kuner[2] & Federico Calegari[1,*] (iD)

## Abstract

Adult neurogenesis is involved in cognitive performance but studies that manipulated this process to improve brain function are scarce. Here, we characterized a genetic mouse model in which neural stem cells (NSC) of the subventricular zone (SVZ) were temporarily expanded by conditional expression of the cell cycle regulators Cdk4/cyclinD1, thus increasing neurogenesis. We found that supernumerary neurons matured and integrated in the olfactory bulb similarly to physiologically generated newborn neurons displaying a correct expression of molecular markers, morphology and electrophysiological activity. Olfactory performance upon increased neurogenesis was unchanged when mice were tested on relatively easy tasks using distinct odor stimuli. In contrast, intriguingly, increasing neurogenesis improved the discrimination ability of mice when challenged with a difficult task using mixtures of highly similar odorants. Together, our study provides a mammalian model to control the expansion of somatic stem cells that can in principle be applied to any tissue for basic research and models of therapy. By applying this to NSC of the SVZ, we highlighted the importance of adult neurogenesis to specifically improve performance in a challenging olfactory task.

**Keywords** adult neurogenesis; neural stem cells; odor discrimination
**Subject Categories** Neuroscience
**The EMBO Journal (2019) 38: e98791**

See also: **M Lipovsek & MS Grubb** (March 2019)

## Introduction

The subventricular zone (SVZ) of the lateral ventricles is the primary neurogenic niche of the adult mammalian brain harboring neural stem cells (NSC) throughout life. This endogenous source of new neurons holds great potential toward therapy, and major efforts are aimed to understand the mechanisms governing the proliferation and differentiation of NSC and the role of adult neurogenesis in cognitive performance and brain plasticity (Silva-Vargas *et al*, 2013; Lim & Alvarez-Buylla, 2014; Lepousez *et al*, 2015; Sailor *et al*, 2017).

Specifically, slowly dividing progenitors of embryonic origin (Fuentealba *et al*, 2015; Furutachi *et al*, 2015) become activated NSC (B1 cells) of the adult SVZ and give rise to intermediate progenitors (C cells) that produce neuroblasts (A cells) migrating through the rostral migratory stream (RMS) and generating interneurons of the olfactory bulb (OB; Doetsch *et al*, 1997). As a result, newborn granule and periglomerular cells are continuously added to the OB, contributing to the plasticity of the local circuitry throughout life by modulating the activity of mitral and tufted cells whose output is projected to the olfactory cortex (Hack *et al*, 2005; Mizrahi *et al*, 2006; Brill *et al*, 2009; Ghosh *et al*, 2011; Miyamichi *et al*, 2011).

Studies attempting to link neurogenesis with olfaction yielded conflicting results pointing to a yet unresolved debate about the role of newborn neurons. For instance, depleting NSC decreased olfactory memory with some (Sakamoto *et al*, 2014) or no (Breton-Provencher *et al*, 2009; Lazarini *et al*, 2009) effect on odor discrimination or learning. Similarly, impairing the migration or survival of newborn neurons had either no effect (Kim *et al*, 2007) or inhibited odor discrimination (Gheusi *et al*, 2000; Bath *et al*, 2008). Seemingly contradicting these findings, promoting neuronal survival also decreased odor discrimination (Mouret *et al*, 2009) or, alternatively, improved learning (Wang *et al*, 2015). Moreover, activating newly integrated cells was reported to facilitate learning and memory by some studies (Alonso *et al*, 2012; Gschwend *et al*, 2015), while, conversely, others showed that enhancing the inhibitory activity of all granule cells in the OB improved odor discrimination time but not learning or memory (Abraham *et al*, 2010; Nunes & Kuner, 2015).

1 CRTD Center for Regenerative Therapies Dresden, School of Medicine, TU Dresden, Dresden, Germany
2 Department of Functional Neuroanatomy, Institute for Anatomy and Cell Biology, Heidelberg University, Heidelberg, Germany
3 Institute of Physiological Chemistry, University Medical Center of the Johannes Gutenberg University Mainz, Mainz, Germany
4 Centre for Developmental Neurobiology and MRC Centre for Neurodevelopmental Disorders, Institute of Psychiatry, Psychology & Neuroscience, King's College London, London, UK
  *Corresponding author. Tel: +49 3514 5882204; E-mail: federico.calegari@tu-dresden.de

These inconsistencies are likely due to the very different approaches previously used by which the intrinsic properties of the mature and/or newborn neurons themselves have been manipulated including at the level of their migration, survival, integration, or electrophysiological properties. In fact, studies investigating whether a specific expansion of NSC resulting in an increased number of otherwise physiologically normal, unmanipulated neurons improves brain function are lacking. Addressing this question is fundamental to identify the specific role of adult neurogenesis in olfaction and explore systems to expand NSC, and perhaps also other somatic stem cells, for therapy.

Our and other groups have shown that the cell cycle regulators Cdk4/cyclinD1 (4D) can be used to regulate the expansion not only of NSC (Lange *et al*, 2009; Artegiani *et al*, 2011; Nonaka-Kinoshita *et al*, 2013) but also human hematopoietic (Mende *et al*, 2015) and pancreatic β-cell (Azzarelli *et al*, 2017; Krentz *et al*, 2017) precursors. Hence, here we decided to develop a versatile transgenic mouse model to temporally control 4D in any tissue of choice. We then used this tool to assess the effects of a cell-intrinsic expansion of adult NSC without a manipulation of their niche and resulting in the increased generation of physiologically normal neurons to study their role in olfactory performance.

## Results

### Temporal control of NSC expansion in the SVZ

Our group has shown that a transient 4D overexpression promoted the proliferation of NSC by shortening their cell cycle, specifically G1. Concomitantly, a shortening of G1 promoted a switch of NSC fate from differentiative to proliferative divisions resulting in the subsequent increase in the number of newborn neurons generated during embryonic cortical development and adult hippocampal neurogenesis (Lange *et al*, 2009; Artegiani *et al*, 2011; Nonaka-Kinoshita *et al*, 2013). Therefore, in order to investigate the effects of increased neurogenesis in odor discrimination, we generated a triple transgenic mouse line by crossing *nestin*^CreERT2 (Imayoshi *et al*, 2008), *ROSA26*^rtTA-flox (Belteki *et al*, 2005), and *tet*^4D-RFP (Nonaka-Kinoshita *et al*, 2013) mice (Fig 1A). This system was designed to allow the tamoxifen (Tam)-dependent activation of rtTA specifically in NSC followed by an inducible and reversible 4D expression, together with RFP as reporter, in a doxycycline (Dox), time-dependent manner. Notably, the *ROSA26*^rtTA-flox/*tet*^4D-RFP line would also allow 4D to be in principle controlled in any other tissue of choice by simply crossing this line with any appropriate Cre driver mouse.

Triple homozygous, adult *nestin*^CreERT2+/+/*ROSA26*^rtTA-flox+/+/ *tet*^4D-RFP+/+ mice (referred to as 4D⁺; see Materials and Methods and Appendix) for the strategy used to obtain this line) were administered Tam for 3 days followed by 4 days of clearance and the subsequent start of Dox administration (defined to as day 0; Fig 1B). After 4 days of Dox, both RFP mRNA and endogenous fluorescence were detected along the SVZ and RMS that, as expected, still did not reach the OB (Fig EV1A). In contrast, neither RFP mRNA nor protein were detectable by *in situ* hybridization or antibody enhancement, respectively, in any other brain area including the hippocampus (Fig EV1A and A'), which is likely due to the lower dosage of Tam relative to that optimized for this niche (Imayoshi

*et al*, 2008; Artegiani *et al*, 2011). No RFP protein could be detected either in the olfactory epithelium (Fig EV1A) underlying the SVZ-specific expression of 4D despite the presence of nestin⁺ cells in other regions of the nervous system.

Within the SVZ, 4D-RFP induction occurred to a similar degree in C and A progenitor cells (72.1 ± 4.8 and 68.2 ± 3.2% of all Mash1⁺ and DCX⁺, respectively) and to a lesser extent in activated B1 cells identified as either EGFR⁺Mash1⁻ or nestin⁺S100β⁻ (Codega *et al*, 2014; 54.4 ± 6.1 and 55.1 ± 2.8%, respectively; Fig 1B). Moreover, RFP mRNA levels were back to undetectable levels 2 days after withdrawing Dox (Fig EV1A'), evidencing the efficiency of our on/off expression system and lack of leakiness.

We next investigated the effects of a 4-day 4D overexpression on proliferation by one pulse of BrdU 12 h before sacrifice (Fig 1C). Hereafter, triple homozygous, *nestin*^CreERT2+/+/*ROSA26*^rtTA-flox+/+/ *tet*^4D-RFP−/− mice (referred to as 4D⁻) in the same genetic background of 4D⁺ mice and equally treated with Tam and Dox were used as negative controls. First, we quantified the proportion of BrdU⁺ cells among activated B1, C, and A cells in 4D⁻ and 4D⁺ mice. We found that in 4D⁺ mice the vast majority (> 80%) of RFP⁺ cells was also BrdU⁺ (Fig 1C). Consistently, the proportion of BrdU⁺ cells among activated B1 cells had substantially increased relative to 4D⁻ mice (from 6.0 ± 0.3 to 42.9 ± 5.9%, $P < 0.005$ and from 28.6 ± 2.1 to 47.7 ± 1.7%, $P < 0.005$, among EGFR⁺Mash1⁻ and nestin⁺S100β⁻ cells, respectively). Note that the different fold-increase by the use of the two marker pairs is likely due to the reported degradation of nestin in S/G2 (Sunabori *et al*, 2008; Codega *et al*, 2014) during which BrdU is incorporated; Fig 1C). A similar increase in the proportion of BrdU⁺ cells was also found among C and A cells in both the SVZ (from 46.9 ± 2.3 to 76.1 ± 5.4%, $P < 0.01$ and from 35.8 ± 3.2 to 69.5 ± 6.4%, $P < 0.01$, for Mash1⁺ and DCX⁺ cells, respectively; Fig 1C) and RMS (from 47.9 ± 2.0 to 78.5 ± 2.1%, $P < 0.01$ and from 39.0 ± 10.5 to 65.4 ± 8.8%, $P = 0.13$, for Mash1⁺ and DCX⁺ cells, respectively; Fig EV1B). These data were consistent with the known effect of 4D in shortening G1 (Lange *et al*, 2009; Artegiani *et al*, 2011; Nonaka-Kinoshita *et al*, 2013) and underlying the observed increase in BrdU incorporation.

Yet, despite the massive increase in BrdU⁺ cells, these results were hard to interpret given that RFP⁺ cells in 4D⁺ mice represented only a fraction (ca. 50%) of all stem and progenitor cells (Fig 1B). Hence, comparison of this subpopulation of RFP⁺ cells in 4D⁺ mice with all cells in 4D⁻ mice might have resulted in a bias if RFP expression was to be enriched in fast-proliferating cells. This was unlikely because BrdU labeling in 4D⁺ mice prior to the beginning of Dox administration, i.e., before a phenotype could be triggered, led to a similar proportion of BrdU⁺ cells among the RFP⁻ and RFP⁺ population (Fig EV1C) indicating that 4D-RFP induction did not bias toward fast-proliferating cells.

Nevertheless, to directly and incontrovertibly exclude the effects of any potential bias in 4D-RFP expression, we quantified the absolute number of BrdU⁺ cells in 4D⁺ and 4D⁻ mice regardless of RFP expression. This was also important because the previous increase in the proportion of BrdU⁺ cells (Fig 1C) primarily reflected a change in cell cycle parameters, but not necessarily in fate, of neural stem and progenitor cells, which could only be proven by observing an increase in their numbers irrespective of BrdU incorporation.

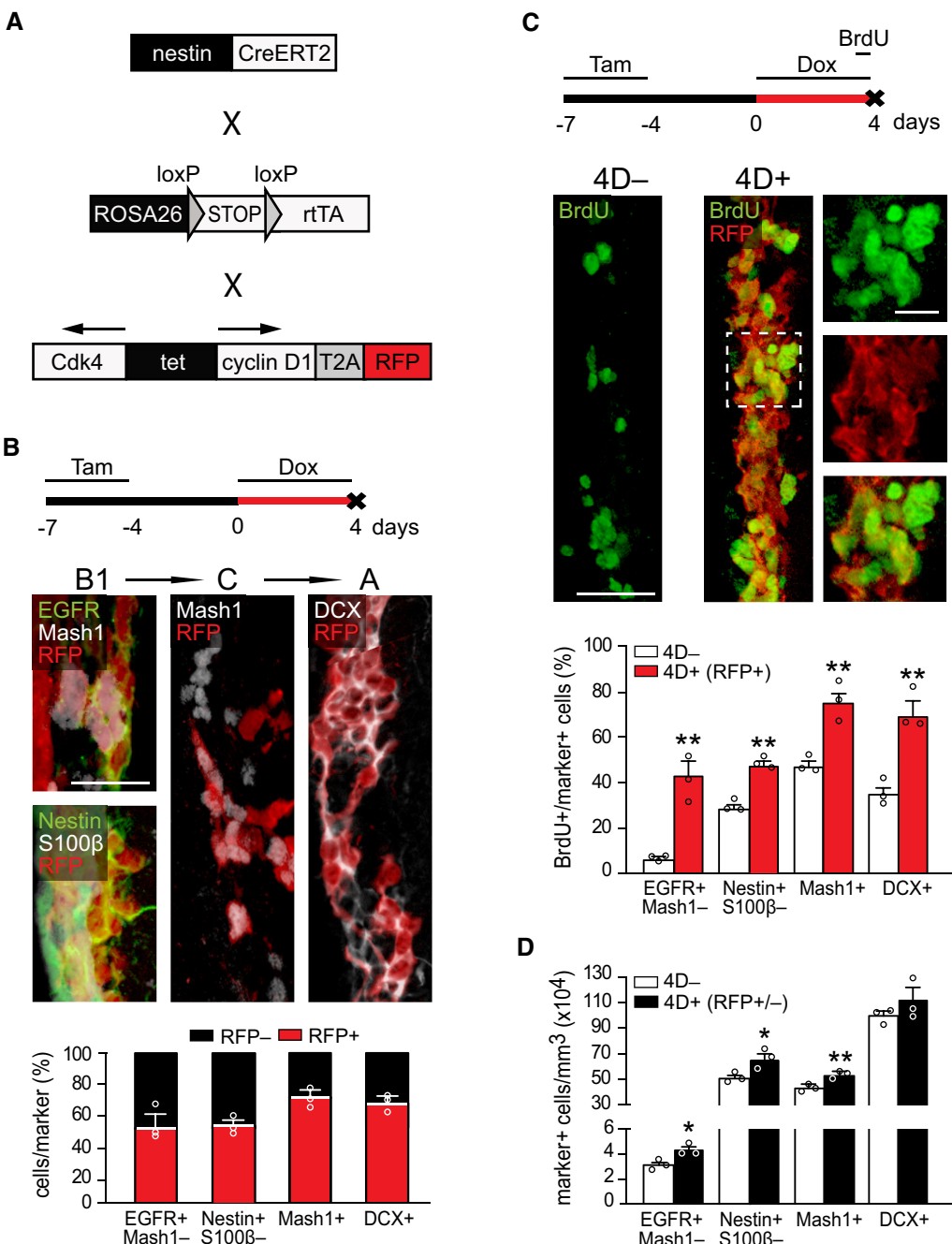

**Figure 1.  Transgenic model and effects of acute 4D overexpression on NSC and progenitors of the SVZ.**

A   Drawings of the *nestin*[CreERT2] (Imayoshi *et al*, 2008), *ROSA26*[rtTA-flox] (Belteki *et al*, 2005), and *tet*[4D-RFP] (Nonaka-Kinoshita *et al*, 2013) alleles of the 4D line.

B   From top to bottom: experimental design of 4D induction, fluorescence pictures of the SVZ of a 4D+ mouse and quantification of the proportion of RFP− (black) and RFP+ (red) progenitors among B1, C, and A cells identified with markers as indicated.

C   From top to bottom: experimental design, fluorescence pictures of the SVZ of a 4D− (left) and 4D+ (right and insets magnified) mice, and quantification of the proportion of BrdU+ among B1, C, and A cells (identified as in B). Note that in 4D+ mice quantification was restricted to the RFP+ subpopulation (red bars).

D   Quantification of the absolute number of B1, C, and A cells (identified as in B) in the SVZ of 4D− (white) and 4D+ (black) mice regardless of RFP expression (RFP+/−).

Data information: (B–D) Mean ± SEM; *$P < 0.05$ and **$P < 0.01$ calculated by unpaired Student's *t*-test; $N = 3$ mice, $n > 423$ cells for each quantification. Scale bar = 50 μm (B and C) or 20 μm (inset in C).

We found that a 4-day induction of 4D triggered an increase by 30% in activated B1 cells per tissue volume (from $3.3 \pm 0.2 \times 10^4$ to $4.4 \pm 0.3 \times 10^4$, $P < 0.05$, and from $51.2 \pm 1.4 \times 10^4$ to $66.6 \pm 4.3 \times 10^4$, $P < 0.05$, EGFR+Mash1− and nestin+S100β− cells/mm³, respectively) and 15% in C cells (from $43.9 \pm 1.6 \times 10^4$ to $52.7 \pm 1.2 \times 10^4$ Mash1+ cells/mm³, $P < 0.01$) throughout the

SVZ (Fig 1D). This increase in cell numbers in 4D$^+$ mice occurred without distinguishing between RFP$^-$ and RFP$^+$ cells indicating that the real 4D-triggered effect on cell fate was greater, in principle the double, than the one assessed.

Altogether, the observed increase in the proportion of BrdU incorporation and number of NSC supports the notion (Lange & Calegari, 2010; Borrell & Calegari, 2014) that 4D overexpression not only induces a faster cell cycle in NSC but also changes their fate by promoting proliferative divisions and expansion of their pool over time.

## 4D expands NSC without inducing their depletion and increases neurogenesis

We next addressed the long-term effect of an acute 4D overexpression. In particular, we investigated whether (i) enhanced NSC proliferation was reversible upon turning off 4D, thus allowing their switch to differentiation; (ii) supernumerary NSC could re-enter quiescence, which is essential to prevent their long-term depletion; (iii) the balance between gliogenic vs. neurogenic commitment was maintained without altering NSC multipotency; and finally, (iv) expansion of NSC increased neurogenesis without compensatory effects. To address all these questions, we designed a common experimental paradigm by which 4D was induced for 4 days concomitantly with BrdU administration followed by a 30- or 60-day chase without Dox (Figs 2A and EV2A).

First, at the end of the described treatment, a single pulse of EdU was given 1 h before sacrifice to investigate whether the 4D effect on the cell cycle and proliferation was reversible (Fig 2A). No difference was found in the number of EdU$^+$ cells ($31.4 \pm 3.2 \times 10^4$ and $33.2 \pm 1.7 \times 10^4$ EdU$^+$ cells/mm$^3$ in 4D$^-$ and 4D$^+$ mice, respectively, $P = 0.66$), indicating that the 4D effect was fully reversible and proliferation was restored back to physiological levels (Fig 2B).

Second, BrdU label retention was used to address entry into quiescence (Doetsch *et al*, 1999). NSC that were cycling during Dox administration and retained the label following a 30-day chase were identified as BrdU$^+$Sox2$^+$S100β$^-$ (Fig 2C; white arrowheads) and found to have increased by twofold in 4D$^+$ mice relative to control ($1.9 \pm 0.3 \times 10^4$ vs. $3.8 \pm 0.3 \times 10^4$ cells/mm$^3$, $P < 0.01$; Fig 2C). This twofold increase in long-term, label-retaining cells seemingly persisted even after a 60-day chase despite the overall age-dependent decrease in the number of quiescent NSC in both cohorts of mice ($1.1 \pm 0.1 \times 10^4$ vs. $1.8 \pm 0.3 \times 10^4$ BrdU$^+$Sox2$^+$S100β$^-$ cells/mm$^3$ in 4D$^-$ and 4D$^+$, respectively, $P = 0.09$; Fig EV2A and B), suggesting a long-term effect by our manipulation without NSC depletion.

Intriguingly, in these experiments quiescent NSC also seemed to appear more frequently as doublets in 4D$^+$ than in 4D$^-$ mice (Fig 2D, arrowheads) suggesting that they were the result of an increase in symmetric, relative to asymmetric, proliferative divisions, both of which are known to occur in the mouse SVZ (Calzolari *et al*, 2015). To investigate this, we took advantage of the fact that while 4D-RFP mRNA expression was terminated soon after Dox removal (Fig EV1A'), RFP as a protein persisted over 1 month later (Fig 2D). Hence, we ranked the proportion of BrdU$^+$RFP$^+$ cells that appeared as doublets (nuclei within 10-μm distance) along the SVZ as a proxy for symmetric divisions, revealing a 30% increase in

4D$^+$ mice as compared to 4D$^-$ (from $51.7 \pm 1.7$ to $67.5 \pm 5.1\%$, $P < 0.05$; Fig 2D) and consistent with the notion that symmetric proliferative divisions underlie the expansion of NSC and their increase in number (Fig 1D).

Third, regarding lineage commitment and multipotency of 4D-expanded NSC, we quantified the proportion of BrdU long-term retaining cells expressing the gliogenic marker S100β and found a similar proportion of mature astrocytes in 4D$^-$ and 4D$^+$ ($43.2 \pm 6.5$ and $48.0 \pm 4.6\%$, respectively, $P = 0.57$; Fig 2C; pie graphs). Hence, as a result of the increased number of NSC and similar proportion of gliogenic commitment, the number of mature astrocytes was also increased in 4D$^+$ mice by twofold ($1.5 \pm 0.1 \times 10^4$ vs. $3.6 \pm 0.8 \times 10^4$ BrdU$^+$S100β$^+$ cells/mm$^3$ in 4D$^-$ vs. 4D$^+$, respectively, $P < 0.05$; Fig 2C; bar graphs). Similar differences were also found after the 60-day chase ($0.9 \pm 0.1 \times 10^4$ vs. $1.3 \pm 0.2 \times 10^4$ BrdU$^+$S100β$^+$ cells/mm$^3$ in 4D$^-$ vs. 4D$^+$, respectively, $P = 0.07$; Fig EV2B).

Finally, we investigated whether the transitorily expanded pool of NSC was capable of undergoing physiological neurogenesis upon silencing of the 4D cassette. Consistent with this, CLARITY treated, whole-mount immunolabeling of 4D$^+$ brains showed widespread distribution of RFP$^+$ cells throughout the entire OB (Movie EV1). Then, we quantified BrdU$^+$ neurons in the OB birthdated during 4D overexpression. In adult mice, most of the newly generated neurons migrating to the OB integrate in the granule cell layer as NeuN$^+$, GABAergic granule cells (Bagley *et al*, 2007; Imayoshi *et al*, 2008; Fig 2E). Additionally, a smaller proportion of GABAergic, glutamatergic or dopaminergic periglomerular interneurons is added to the glomerular layer that can be classified in three mutually exclusive populations of calretinin$^+$ (CalR), calbindin$^+$ (CalB), or tyrosine hydroxylase$^+$ (TH) cells (Hack *et al*, 2005; Parrish-Aungst *et al*, 2007; Brill *et al*, 2009). We observed an evident increase in BrdU$^+$ neurons in 4D$^+$ mice (Fig 2E) that contributed to the granule cell and glomerular layers in proportions similar to that observed in 4D$^-$ mice ($82.6 \pm 1.3$ and $17.4 \pm 1.3\%$ vs. $82.8 \pm 2.2$ and $17.2 \pm 2.2\%$, for granule cell and glomerular layers, in 4D$^+$ and 4D$^-$ mice, respectively, $P = 1.0$; Fig 2E; pie graphs). Regarding absolute numbers, 30 days after 4D overexpression we observed a similar increase in all neuronal types that reached our threshold of statistical significance for both NeuN$^+$ granule cells (from $4.42 \pm 0.46 \times 10^4$ to $6.05 \pm 0.36 \times 10^4$ cells/mm$^3$, $P < 0.05$) and TH$^+$ periglomerular cells (from $0.23 \pm 0.03 \times 10^4$ to $0.44 \pm 0.07 \times 10^4$ cells/mm$^3$, $P < 0.05$; Fig 2E). Similar differences were found after the 60-day chase, time at which also the increase in CalB$^+$ cells reached statistical significance (Fig EV2C).

Together, these data outline the effects of 4D on NSC resulting in their long-term expansion and increased neurogenesis without affecting their multipotency or compensatory effects due to depletion and/or neuronal death.

## 4D expression in NSC does not alter the morphology or activity of supernumerary neurons

We next assessed whether the integration and activity of supernumerary neurons was altered by the nature of our manipulation in progenitor cells. To this aim, we focussed on granule cells since they represent the most abundant type of adult born neurons playing key roles in olfactory discrimination (Abraham *et al*, 2010; Alonso *et al*,

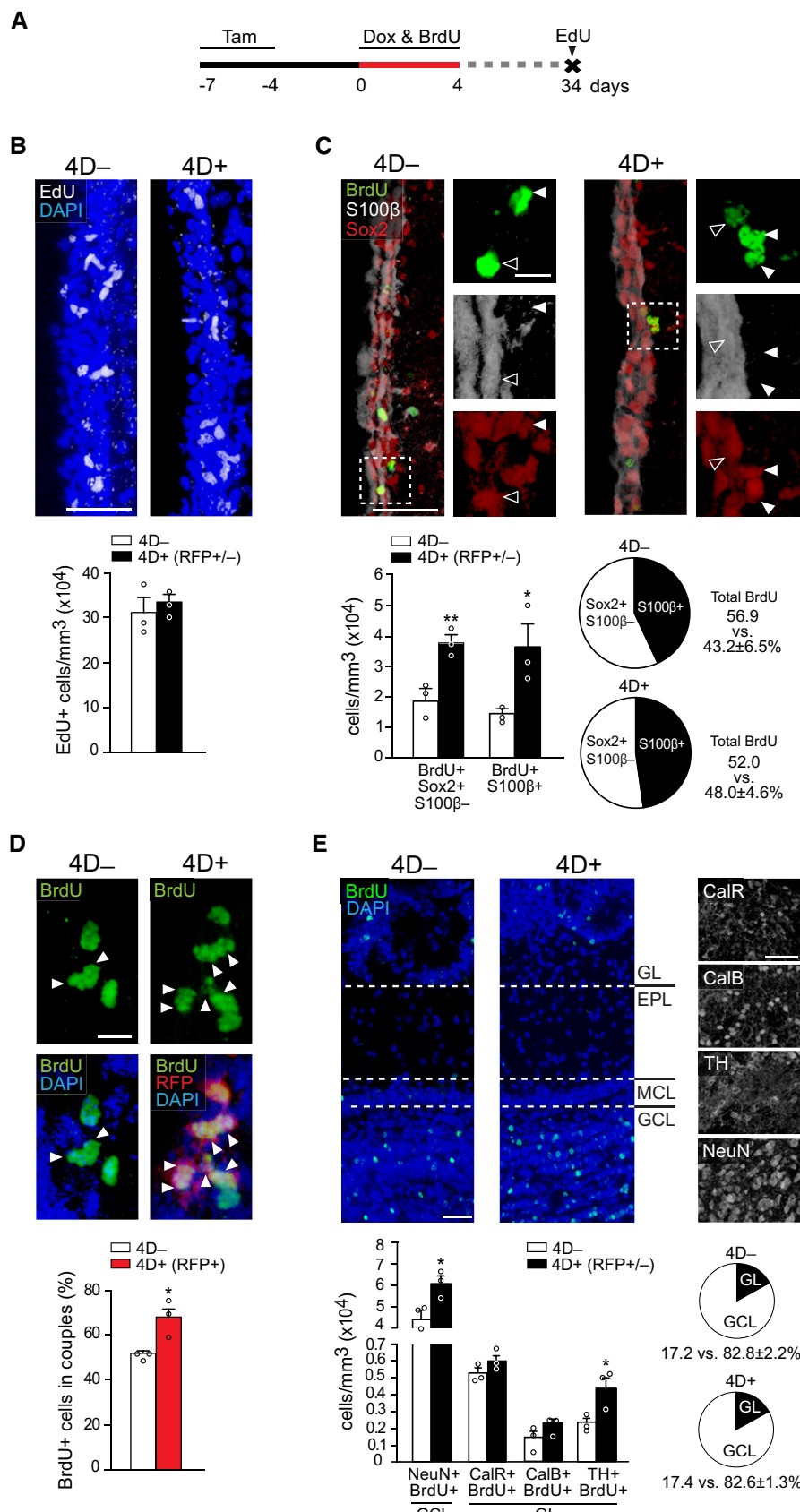

**Figure 2.**

◄

**Figure 2. Chronic effect of 4D overexpression on NSC and OB neurogenesis.**

A    Experimental design used to assess the chronic effect of a transient 4D induction with BrdU and EdU given during Dox administration or 1 h before sacrifice, respectively.

B–E    From top to bottom: fluorescence pictures of the SVZ (B–D) or OB (E) and absolute number (B, C, and E) or proportions (C–E) of cells in 4D$^-$ (white bars) or 4D$^+$ (black or red bars for all or RFP$^+$ cells, respectively) mice scored positive for various markers as indicated. Insets in (C) are magnified (right) with arrowheads pointing label-retaining NSC (white) or astrocytes (empty). Arrowheads in (D) point cell doublets (among RFP$^+$ protein-retaining cells in 4D$^+$). (E) GL, glomerular; EPL, external plexiform; MCL, mitral cell and GCL, granule cell layers.

Data information: (B–E) Mean ± SEM; *$P < 0.05$, **$P < 0.01$ assessed by unpaired Student's *t*-test (bar graphs) or Fisher's exact test (pie graphs); $N = 3$ mice, $n > 285$ cells for each quantification. Scale bars = 50 μm (B, C, and E) and 20 μm (D and insets in C).

2012; Gschwend *et al*, 2015; Nunes & Kuner, 2015). NSC expansion was induced for 4 days and granule cells in the OB analyzed 30 days later (Fig 3A), time at which adult born neurons are known to be morphologically mature and integrated (Petreanu & Alvarez-Buylla, 2002). Here, in contrast to our previous quantifications of cell numbers alone (Figs 1 and 2), we needed a system that would allow us to identify supernumerary RFP$^+$ neurons in 4D$^+$ mice and compare them with physiologically generated, RFP$^-$ newborn neurons of an equivalent age in 4D$^-$ or even within 4D$^+$ mice. To mark such age-matched cohort of newborn neurons in 4D$^-$ and 4D$^+$ mice, we then crossed the homozygous 4D$^-$ and 4D$^+$ lines with $RCE^{GFP\text{-}flox+/+}$ mice (Miyoshi *et al*, 2010), thus, labeling *nestin*-CreERt2+ NSC upon Tam administration by GFP, irrespective from the presence or absence of RFP. We then compared superficial granule neurons derived from 4D$^+$ NSC (RFP$^+$GFP$^+$) with physiologically generated neurons of the equivalent age but derived from 4D$^-$ NSC (RFP$^-$GFP$^+$).

Morphometric and 3D-Sholl analyses revealed that spine density (0.36 ± 0.02 vs. 0.33 ± 0.03 spines/μm, 4D$^-$ and 4D$^+$, respectively, $P = 0.44$), total dendritic length (0.78 ± 0.6 and 0.66 ± 0.8 mm, $P = 0.26$), and arborization (intersections as a function of distance from the soma $F_{(1,28)} = 0.61$, $P = 0.44$) were comparable in granule cells derived from 4D$^-$ and 4D$^+$ NSC (Fig 3B) and fitting well with previous reports (Abraham *et al*, 2010; Scotto-Lomassese *et al*, 2011; Breton-Provencher *et al*, 2016). 4D-derived granule neurons expressed the presynaptic vesicular GABA transporter VIAAT that co-localized with the post-synaptic marker gephyrin (Nunes & Kuner, 2015; Fig 3C). Moreover, characteristic synaptic clefts and vesicles

were observed in 4D-derived neurons at the ultrastructural level by electron microscopy upon RFP immunogold labeling (Fig 3D), thus evidencing the presence of mature synapses.

To further confirm the functional integration of 4D-derived neurons, we next assessed their electrophysiological properties. Patch-clamp recordings in the OB were performed on slices from 4D$^-$ and 4D$^+$ mice comparing newborn granule neurons identified by GFP and/or RFP expression as described above. This showed that both cohorts of neurons received spontaneous barrages of synaptic input of similar frequency and amplitude (values for 4D$^-$ vs. 4D$^+$, respectively: 2.9 ± 0.8 vs. 3.6 ± 0.8 Hz, $P = 0.52$; 2.6 ± 0.5 vs. 2.7 ± 0.7 mV, $P = 0.87$; Figs 3E and EV3B). Properties of action potentials in response to depolarizing current steps were also similar as well as input resistance and resting membrane potential (spike number: 12.5 ± 0.9 vs. 13.7 ± 2.8 spikes/500 ms, $P = 0.71$; spike width: 1.7 ± 0.1 vs. 1.4 ± 0.1 ms, $P = 0.12$; spike amplitude: 66.7 ± 3.5 vs. 72.3 ± 4.8 mV, $P = 0.35$ and after hyperpolarization: 38.1 ± 3.9 vs. 43.1 ± 2.3, $P = 0.33$; R$_{in}$: 892.7 ± 86.6 vs. 956.8 ± 103.7 MΩ, $P = 0.64$ and V$_{rest}$: −66.0 ± 3.4 vs. −60.7 ± 4.2 mV, $P = 0.33$; Figs 3F and H, and EV3B). We also compared several other parameters including capacitance, voltage threshold for spike initiation, and rheobase, which in all cases yielded expected, and virtually identical, values between age-matched cohorts of physiologically generated and 4D-derived neurons (Fig EV3B). Finally, both cohorts of neurons displayed the characteristic lag in the initiation of the first action potential in response to prolonged current injection (78.5 ± 12.2 vs. 79.5 ± 12.3 ms, $P = 0.95$; Fig 3G and H) consistent with the

**Figure 3. Integration and electrophysiological properties of 4D-derived granule cells.**                                    ►

A    Experimental design to assess the integration of 4D-derived neurons.

B    Fluorescence pictures (left) of immunolabeled GFP$^+$RFP$^-$ (4D$^-$) and GFP$^+$RFP$^+$ (4D$^+$) apical dendrites of superficial granule neurons and quantifications (middle and right) of spine density and total dendritic length (box and whiskers) and 3D-Sholl (line graph profile) of apical dendrites starting from the soma (drawings) as the mean number of intersections at 10-μm intervals.

C    3D reconstruction of multi-channel confocal stacks acquired from a RFP$^+$ apical dendrite of a granule cell showing co-localization with the pre- and post-synaptic markers VIAAT (granule neuron) and gephyrin (mitral cell), respectively (magnification shown in insets).

D    Anti-RFP immunogold labeling of a cell in the superficial granule cell layer of a 4D$^+$ mouse. Inset shows a representative RFP$^+$ synapse (out of > 10 analyzed from 4D$^+$ mice).

E–G    Current clamp recordings showing examples of spontaneous barrages of synaptic potentials (E), and repetitive spiking in response to depolarizing current steps (F, G) of 4D$^-$ (top) and 4D$^+$ (bottom) mice. Inset in E (4D$^+$ cell) is magnified (bottom). Note in (G) that the lag preventing spike initiation is longer at lower currents (green) and shorter at higher (black), suggesting the presence of an A-type K current typical of granule cells in 4D$^-$ vs. 4D$^+$.

H    Box and whiskers plots representing electrophysiological properties of neurons derived from 4D$^-$ and 4D$^+$ NSC (black and red, respectively) including from left to right: spike number, resting membrane potential (Vrest), input resistance (Rin), and lag to spiking of the recorded superficial granule cells (see Fig EV3 for additional parameters).

Data information: Data are presented as mean ± SEM in the line graph in panel (B). No significant difference was found by unpaired Student's *t*-test (throughout) or repeated measures two-way ANOVA (line graph in B). Boxplots in (B and H) show the median (horizontal line), and mean (+) and whiskers indicate the lowest and highest values within 1.5 interquartile range. Outliers were identified by Tukey's test. (B) $N = 3$ mice, $n > 8$ neurons per genotype; (E–H) $N > 5$ mice, $n = 12$ 4D$^-$ and 10 4D$^+$ neurons. Scale bars = 5 μm (B and C), 1 μm (D and insets in C), and 0.5 μm (inset in D).

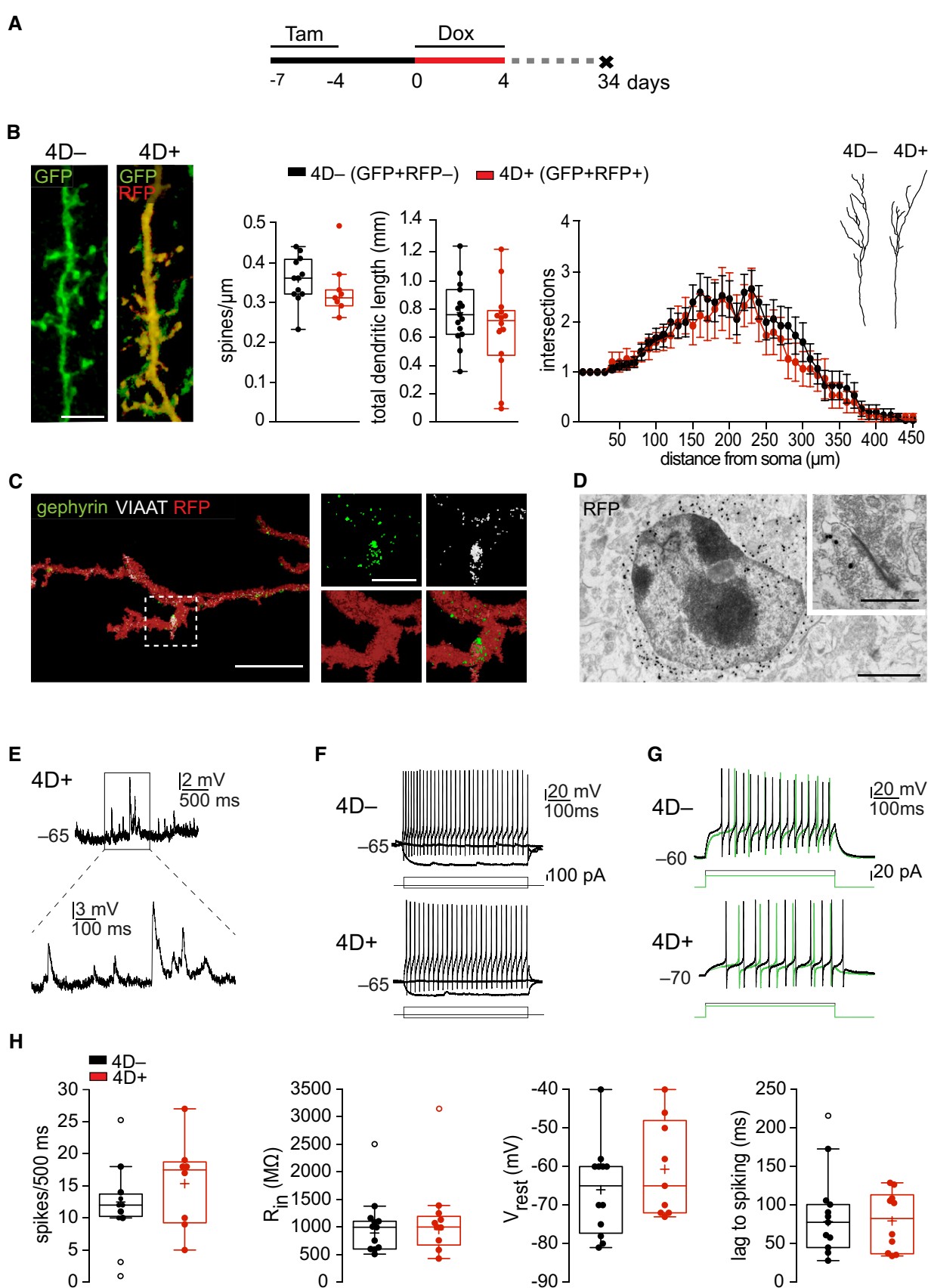

Figure 3.

presence of A-type $K^+$ currents typical of granule cells (Schoppa & Westbrook, 1999).

In summary, both morphometric and electrophysiological analyses (Figs 3 and EV3), together with our previous assessment of molecular markers (Fig 2), confirmed that the maturation, integration, and activity of 4D-derived granule cells were in all aspects similar to that of physiologically generated neurons. This contrasts the differences observed in endogenous vs. graft-derived interneurons following transplantation of neural precursors (Larimer et al, 2017) suggesting that supernumerary neurons by our 4D manipulation in NSC not only are similar in cell-intrinsic properties but also have no competitive disadvantage compared to endogenous neurons. This in turn distinguishes our approach from previous studies assessing the role of neurogenesis in olfaction upon manipulations changing the intrinsic properties of the neurons themselves and/or their niche.

## Increased neurogenesis improves discrimination accuracy of highly similar odors

To investigate the potential contribution of the supernumerary neurons on olfaction, we treated mice as previously described (Fig 3A) followed by a go/no-go odor discrimination task (Fig 4A). In particular, mice were exposed to either of two odorants and learned to discriminate between the unrewarded and rewarded stimulus after which mice are expected to remove their head from the sampling port or keep it in and lick for water, respectively (Abraham et al, 2010; Nunes & Kuner, 2015; Fig 4B). This allowed the assessment of learning performance calculated as the percentage of correct responses over testing as well as the discrimination time (DT) defined as the time needed to decide how to respond to the odorants at maximal accuracy (referred to as criterion; see Materials and Methods and Appendix).

Mice were serially exposed to discrimination tasks of increasing difficulty. To start with, amyl acetate (AA) and ethyl butyrate (EB) were used as pure odorants and, sequentially, as binary mixtures at 40/60% vs. 60/40%. In either case, a similar learning performance (group $F_{1,36} = 1.09$ and $1.96$, $P = 0.30$ and $0.17$ for pure odors and binary mixtures, respectively) was found in $4D^-$ and $4D^+$ mice (Fig EV4B). We then used two enantiomers of the same odorant, $(-)$ or $(+)$-octanol, as a more difficult task but again no difference was found neither using pure odors (group $F_{1,35} = 0.001$, $P = 0.97$) nor their binary mixtures (group $F_{1,33} = 0.24$, $P = 0.63$; Figs EV4B and 4C). Furthermore, no difference was observed in DTs of $4D^-$ and $4D^+$ mice in any of these discrimination tasks (Fig EV4C).

However, in all these tasks mice typically reached the criterion (> 95% performance) within the first 200–500 trials and maintained this high performance throughout the testing. This raised the possibility that the stimuli were too easy to discriminate and that the task could be efficiently performed independently from any potential gains that an increase in the number of newly integrated granule cells could provide. In other words, the stimuli tested so far might not have pushed the olfactory system into the critical range of performance within which differences in the number of newborn granule cells may become physiologically relevant.

To address this, we subjected the mice already tested on the binary mixture of enantiomers (Fig 4C) to the same task but this time with odorants diluted 1:10 as a probe trial after that a similar maximal performance was reached in the previous task. Notably, this mixture and dilution of octanols was intentionally chosen as one of the most challenging olfactory tasks that mice can perform but still at a dilution range above the detection threshold (preprint: Abraham et al, 2018) to specifically assess discrimination performance rather than detectability. As an evidence of the difficulty of this task, both groups of mice for the first time failed to reach the previous 95% performance and remained at a lower efficiency of about 70%. In this probe test, we found that $4D^+$ mice showed a significantly better discrimination accuracy that increased their performance relative to $4D^-$ controls from $65.8 \pm 2.9$ to $72.1 \pm 3.2\%$ ($P < 0.05$; Fig 4D). This increased performance of $4D^+$ mice was maintained throughout the testing with no obvious effect over time (Fig EV4B; right). Moreover, when assessing the proportion of trials in which mice licked to receive water irrespective of the odorant being presented, we found that control mice licked more often than $4D^+$ ($65.3 \pm 3.3$ and $49.0 \pm 4.8$, $P < 0.01$; Fig 4D). In essence, this suggested that during the diluted octanol trials the lower performance of control mice induced them to lick more often to compensate for the average decrease in total water obtained relative to $4D^+$.

Altogether, these data highlight the effect of an additional supply of newborn neurons in improving olfactory accuracy and increasing discrimination performance specifically in an extremely challenging task.

# Discussion

Previous studies reported that an expansion of NSC in the adult hippocampus was sufficient to improve cognitive performance (Cao et al, 2004; Sahay et al, 2011; Stone et al, 2011). However, experiments establishing a link between NSC expansion and brain function in the second and major adult neurogenic niche of the SVZ are lacking.

Here, we reported a system to expand long-term multipotent NSC of the adult SVZ by an increase in symmetric proliferative divisions and without inducing their depletion. Moreover, the transient nature of our manipulation allowed us to switch this expanded pool of NSC back to neurogenesis resulting in an increase in newborn neurons integrating in the brain circuitry and preserving a seemingly normal expression of molecular markers, morphology and electrophysiological properties. As a result, this led to an improvement in odor discrimination accuracy specifically when mice were challenged by highly similar odorants. Several observations of our study are worth discussing.

First, validating the notion that a short cell cycle acts as a determinant of somatic stem cell fate (Lange & Calegari, 2010; Borrell & Calegari, 2014), 4D was originally shown to promote the expansion of NSC in the developing and adult brain (Lange et al, 2009; Artegiani et al, 2011; Nonaka-Kinoshita et al, 2013) and more recently also of human hematopoietic stem cells upon bone marrow transplantation (Mende et al, 2015), pancreatic β-cell precursors in models of diabetes (Azzarelli et al, 2017; Krentz et al, 2017), and cardiomyocytes during heart regeneration (Mohamed et al, 2018). Thus, on a purely technical ground, our $ROSA26^{\text{rtTA-flox}+/+}$ / $tet^{\text{4D-RFP}+/+}$ mouse line, after crossing with any appropriate Cre line of choice, offers the unique possibility to temporarily control the

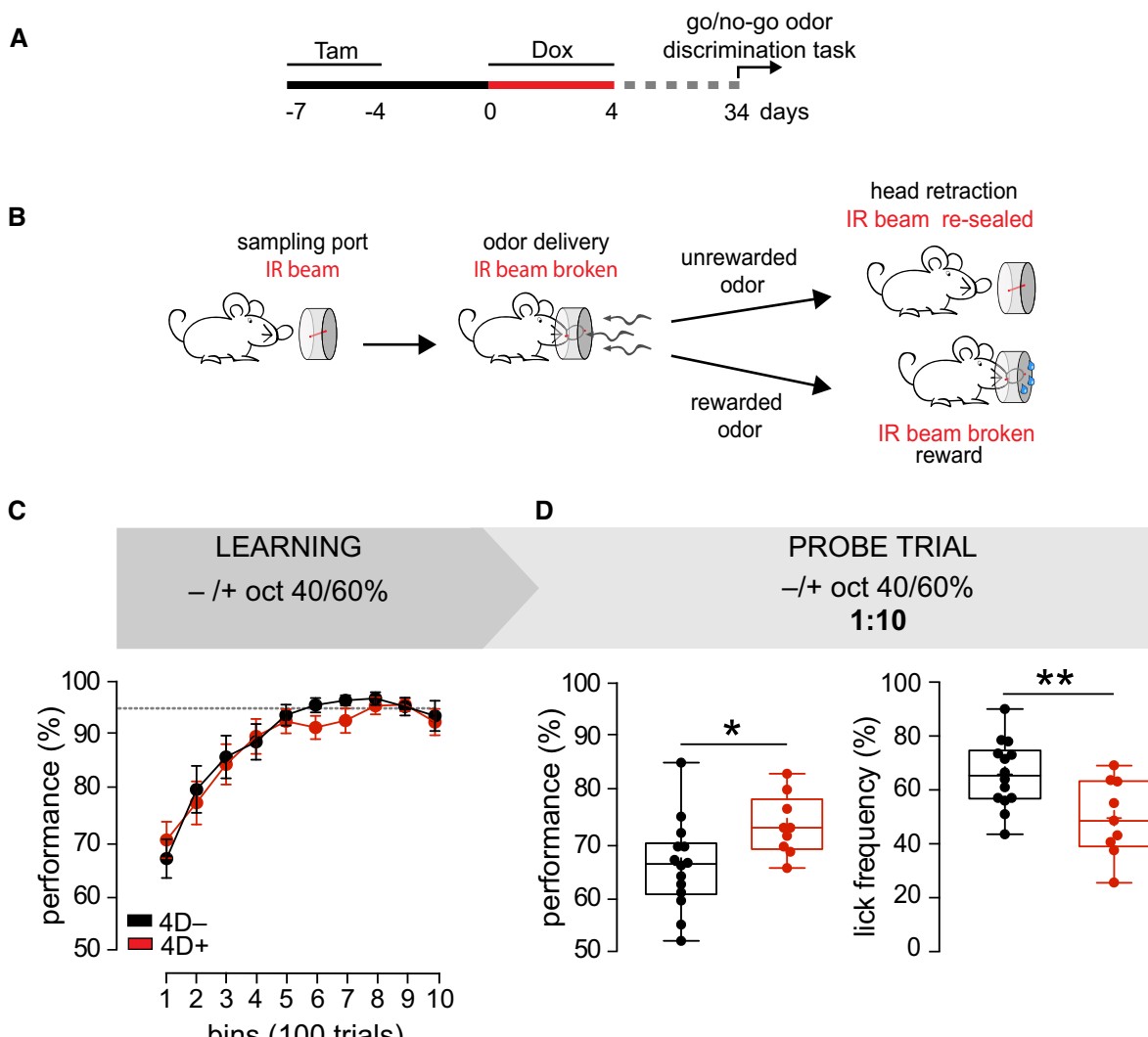

**Figure 4.  Effect of increased neurogenesis on olfaction.**

A   Experimental design to test effects on olfaction.

B   Scheme of the go/no-go odor discrimination task. Mice introduce their head in the sampling port breaking the infrared beam (IR). A correct response is scored when mice retract their head in the presence of an unrewarded odor or, alternatively, wait and lick for a rewarded one. See additional information in Appendix.

C   Line graph indicating the proportion of correct responses (performance) for bins of 100 trials each during testing with binary mixtures of concentrated octanols. Discontinuous line indicates the similar 95% plateau performance of 4D⁻ (black) and 4D⁺ (red) mice.

D   Box-and-whisker plots of performance (left) and lick frequency (right) during a probe of 200 trials with 1:10 diluted octanols after testing as in (C).

Data information: (C) Mean ± SEM with significance assessed by repeated measures two-way ANOVA. (D) Unpaired Student's *t*-test *$P < 0.05$ and **$P < 0.001$. Boxplots show the median (horizontal line) and mean (+). Whiskers indicate the lowest and highest values within 1.5 interquartile range. Cohorts of $N$ = 21/17 (C) or 15/10 (D) 4D⁻/4D⁺ mice were used.

expansion of, in principle, any somatic stem cell as a powerful new approach for basic research and experimental models of regenerative therapy.

Second, many studies investigated the role of adult SVZ neurogenesis by depleting NSC and/or altering the properties of the neurons themselves at the level of their migration, survival, integration, and/or electrophysiological activity (Mouret *et al*, 2009; Abraham *et al*, 2010; Alonso *et al*, 2012; Gschwend *et al*, 2015; Nunes & Kuner, 2015; Wang *et al*, 2015). While this led to several conflicting results and a controversy in the field, our approach was fundamentally different with the transient nature of our manipulation

specifically within NSC ensuring that the supernumerary neurons were not affected by ectopic gene expression, hence, preserving their intrinsic physiological properties and true function. This was particularly the case by using 4D. In fact, not only were the two functional transgenes Cdk4 and cyclinD1 transitorily expressed solely in NSC but also cyclinD1 is known to be degraded before entry in S phase, hence, preventing the inheritance of any residual protein by the neuronal progeny. As a result, the cyclinD1-dependent activity of Cdk4 ensures that also this second transgene would become biologically inactive in newborn neurons. Consistently, no detectable difference was found in 4D-derived, supernumerary

neurons neither at the level of their migration, survival, integration, maturation, and including the expression of several molecular markers, their morphological properties and electrophysiological activity.

In these settings, third, an increase in newborn neurons induced no visible effect in olfaction in easy discrimination tasks with both control and $4D^+$ mice reaching a similar maximum performance of nearly 100%. Yet, a significant improvement was observed when the task was made sufficiently difficult to reduce performances to only about 70%. Notably, such improvement appeared despite the fact that several limitations contributed to a substantial underestimation of the phenotypes assessed. Among them, in this study 4D was activated only within a subpopulation of about 50% of NSC implying that the effects could have been greater, in principle the double, than the ones assessed. More importantly, when examining the number of newly integrated granule cells, we found in $4D^+$ mice an increase by approximately 40% relative to controls. Yet, when extrapolating the relative contribution of this subpopulation of newborn neurons as a proportion of the total number of all mature granule cells in the OB (ca. 15%; Ninkovic et al, 2007), this 4D-induced increase represented only a few percent (ca. 5%) of the total. This has the important implication that even a relatively small addition of newborn neurons was sufficient to trigger a significant effect on brain function provided that the task was made sufficiently difficult to make such a small increase functionally relevant. This in turn raises new questions for future studies with regard to further potential gains in brain function(s) that were not observed, or simply not addressed, in our current study should our manipulation be optimized to increase neurons beyond what currently achieved. These may include new potential gains in learning, DT and/or switching the physiological threshold of detection or discrimination in new tasks with exceptionally diluted odorants.

Finally, fourth, NSC of embryonic origin populate specific areas the postnatal brain (Fuentealba et al, 2015; Furutachi et al, 2015) and generate neurons that in humans integrate in the neonatal frontal lobes (Sanai et al, 2011; Paredes et al, 2016) and adult striatum (Ernst et al, 2014). As a result, a number of sensorimotor, cognitive, and neurodegenerative diseases, among others epilepsy, autism, and Huntington's, are thought to involve postnatal neurogenesis (Ernst et al, 2014; Paredes et al, 2016). Our finding that expansion of NSC and an overall relatively minor increase in the total number of neurons improved discrimination performance in physiological conditions provides a proof-of-principle and potential approach toward exploring the use of endogenous NSC to promote recovery of brain function in aging or disease.

# Materials and Methods

### Animals and treatments

Triple transgenic $4D^-$ and $4D^+$ mice were generated starting from the individual heterozygous lines previously described (Nonaka-Kinoshita et al, 2013). After genetic background homogenization for > 5 generations, triple homozygous $4D^-$ and $4D^+$ were selected as founders that were eventually crossed with $RCE^{GFP-flox+/+}$ mice (Miyoshi et al, 2010) for morphometrical and electrophysiological measurements (see Appendix for additional information about the strategy used to obtain these lines). Tamoxifen (Sigma) was

administered at 250, 9-tert-butyl doxycycline (Echelon Biosciences) at 50, BrdU (Sigma) at 50, and EdU (Sigma) at 5 mg/kg body weight (see Appendix). Mice were anesthetized with pentobarbital and perfused transcardially with PBS followed by 4% PFA fixation. Animal procedures were approved by local authorities (DD24-9168.11-1/2011-11, TVV13/2016, and HD35-9185.81/G-61/15).

### Immunohistochemistry and in situ hybridization

Perfused brains were post-fixed overnight in 4% PFA at 4°C. For histology, 40-μm-thick vibratome sections were stored at −20°C in cryoprotectant solution (25% ethylene glycol and 25% glycerol in PBS). Immunohistochemistry was performed as described (Artegiani et al, 2011). Briefly, blocking and permeabilization were performed in 10% donkey serum, 0.3% Triton X-100 in PBS for 1 h at RT, and antibodies (Table 1) diluted in 3% donkey serum, 0.3% Triton X-100 in PBS and incubated overnight at 4°C. For BrdU detection, sections were incubated with HCl 2 M for 25 min at 37°C. Eventually, click reaction was performed for EdU detection (Life Technologies). DAPI was used to visualize nuclei. For whole-mount immunostaining, a modified version of the iDISCO method was used (Renier et al, 2014; see Appendix). RFP in situ hybridization was performed as described (Nonaka-Kinoshita et al, 2013; see Appendix).

**Table 1.  List of antibodies.**

| Primary antibodies | | | |
|---|---|---|---|
| BrdU | 1:250 | Abcam | ab6326 |
| CalB | 1:250 | Swant | 300 |
| CalR | 1:250 | Swant | CR 7697 |
| DCX | 1:100 | Santa Cruz Biotechnology | Sc-8066 |
| Digoxigenin | 1:5,000 | Roche | 11093274910 |
| EGFR | 1:250 | Novus Biologicals | nb110-56945 |
| Gephyrin | 1:300 | Synaptic Systems | 147 021 |
| GFP | 1:400 | Rockland | 600-101-215 |
| Mash1 | 1:100 | BD Biosciences | 556604 |
| NeuN | 1:250 | Millipore | MAB377 |
| RFP | 1:2,000 | Rockland | 200-301-379 |
| S100β | 1:1,000 | Abcam | ab14688 |
| TH | 1:250 | Millipore | MAB5280 |
| VIAAT | 1:400 | Synaptic Systems | 131 004 |

| Secondary antibodies | | | |
|---|---|---|---|
| Alexa Fluor® AffiniPure | 1:1,000 | Jackson ImmunoResearch Laboratories | |
| F(ab')2 GAR Ultra Small | 1:30 | Aurion | 800–166 |

From left to right: antigen recognized, dilution used, provider and catalog number of primary (top) and secondary (bottom) antibodies used.

**Table 2.  List of odorants.**

| Odorants | | |
|---|---|---|
| Cineole (Cin) | Sigma | #27395 |
| Eugenol (Eu) | Fluka | #46100 |
| Amyl acetate (AA) | Sigma | #109584 |
| Ethyl butyrate (EB) | Sigma | #E15701 |
| (+)-Octanol (+)-Oct | Fluka | #74863 |
| (−)-Octanol (−)-Oct | Fluka | #74865 |

From left to right: odorant name, provider and catalog number of odorants used.

**Electron microscopy**

Immunogold labeling was performed as described (Kurth *et al*, 2010; see Appendix and Table 1), and samples analyzed on a Morgagni D268 (FEI) or a JEM1400 Plus (JEOL) at 80 kV acceleration voltage.

**Image acquisition and cell quantification**

Immunohistochemistry, *in situ* hybridization, and clarity images were acquired with an automated Zeiss ApoTome, confocal microscope (LSM 780, Carl Zeiss) and Ultramicroscope (LaVision BioTec, Germany), respectively (see Appendix). For cell quantification, stereological analysis was performed using 1 every six sections from the SVZ and RMS or 1 every three from the OB. For Sholl analyses, z-stacks separated by 1 μm were 3D reconstructed and dendrites traced using the Fiji plug-in Simple Neurite Tracer and radii of 10 μm.

**Electrophysiology**

300-μm-thick vibratome OB slices were used for patch-clamp whole-cell recordings using an Axopatch 200B, pClamp10 (Molecular Devices) for generating current steps and Clampfit for data analysis (see Appendix for a detailed description of electrophysiological measurements).

**Olfactometry**

Behavioral tests were performed by an experimenter blind to the manipulation using a go/no-go operant conditioning scheme (Abraham *et al*, 2004) in a fully automated, custom-made olfactometer in which non-olfactory cues were previously assessed and excluded (Appendix). Eight-week-old males were individually marked by a transponder and several parameters assessed during testing, including body weight, licking frequency, circadian rhythms, and others and in which no differences appeared during the course of the tests and/or $4D^-$ vs. $4D^+$ mice (Appendix). Odors (Table 2) were dissolved in mineral oil at a final concentration of 1%. Under these conditions, trained mice retracted their heads from the sampling port when unrewarded odorants were presented or, alternatively, kept their heads inside when facing the rewarded odorant until presentation was completed (2 s) and starting to lick to receive water. Performance was calculated as the percentage of correct responses (go/no-go and lick) in bins of 100 trials (200 for the probe test). Only mice completing at least 1,000 trials were considered for analysis. Correct trials upon reaching criterion (95% performance) were used to calculate the DT (see Appendix; Abraham *et al*, 2010).

**Statistical analyses**

Data were reported as mean ± SEM. Significance was calculated by two-tailed, unpaired Student's *t*-test assuming unequal variance throughout except for the use of Fisher's exact test for comparing parts of the whole (Figs 2 and EV2) and repeated measures, two-way ANOVA for Sholl analyses (Fig 3) and performance in olfaction (Figs 4 and EV4). Morphometric analysis, electrophysiology, discrimination performance, and licking frequency during the probe test and DTs were represented as whiskers box plots with outliers identified by Tukey's test (Figs 3, 4, EV3, and EV4). Number of biological replicates, either animals (*N*) and/or cells (*n*), used for quantifications of each experiment were indicated in the respective figure legends.

**Expanded View** for this article is available online.

## Acknowledgements

This work was supported by the TUD-CRTD (FZT111) and the DFG (CA 893/6-1 and 16-1 to FC; SPP1392 to TK, INST 161/875-2 to BB). N.M. was supported by a fellowship from the Human Frontiers Science Program (HFSP Long-Term Fellowship, LT000646/2015). We are grateful to Dr. Schaefer A for providing the blueprint of an early prototype of the automated olfactometer, Dr. Kurth T (CRTD) for electron microscopy data, IMBI Heidelberg for help with statistical analyses, Dr. Garthe A (DZNE) for helpful discussion, Rodriguez Cabrera LA for whole mount images, the CRTD and MPI-CBG animal houses and DZNE and CRTD LMF facilities for support.

## Author contributions

SBA and FC conceived the project, designed the experiments, interpreted the data and wrote the manuscript. SBA performed all experiments with the support of SM except for olfactometry performed by JKR and TK and electrophysiology by NM and BB. All authors contributed to and approved the manuscript.

## Conflict of interest

The authors declare that they have no conflict of interest.

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
