## [Review Process File · The EMBO Journal]

An Increase in Neural Stem Cells and Olfactory Bulb Adult Neurogenesis Improves Discrimination of Highly Similar Odorants

Sara Bragado Alonso, Janine K. Reinert, Nicolas Marichal, Simone Massalini, Benedikt Berninger, Thomas Kuner and Federico Calegari

Review timeline:

Submission date:	8th Dec 2017
Editorial Decision:	12th Feb 2018
Revision received:	28th Sep 2018
Editorial Decision:	24th Oct 2018
Revision received:	14th Nov 2018
Accepted:	15th Nov 2018

Editor: Karin Dumstrei

Transaction Report:

1st Editorial Decision

12th Feb 2018

Thank you for submitting your manuscript to The EMBO Journal. I am sorry for the delay in getting back to you with a decision, but I have now received the input from the three referees.

As you can see from the comments below, the referees find the analysis interesting but it is also clear that more work is required for consideration here. The referees find that further data is needed to support that the observed increased adult neurogenesis leads to improved odor discrimination. The referees offer a number of constructive comments for how to strengthen this part. Should you be able to address the raised concerns then I would like to invite you to submit a revised version. I can extend the revision time up to 6 months if that is helpful.

Let me know if we need to discuss anything further.

REFeree REPORTS:

Referee #1:

The authors examine the effects of induced neural stem cell expansion in the SVZ on OB neurogenesis and function. The experiments are well executed and the findings are consistent with

previous studies (using optical stimulation and LOF approaches) implicating OB neurogenesis in discrimination of similar odor stimuli. The authors provide compelling evidence for a powerful way to amplify the population of neural stem cells *in vivo* to enhance neurogenesis. I think it would strengthen the study if the authors can provide some evidence for changes in downstream circuits like the Piriform cortex in these mice. One way to assess this is by c-Fos immunohistochemistry in 4D⁺ and 4D⁻ mice following discrimination.

Referee #2:

In this manuscript, Bragado Alonso and colleagues use mouse genetics to selectively target Cdk4/cyclinD1 (4D) overexpression to neural stem cells of the subventricular zone (SVZ). The authors find that transient 4D overexpression leads to a shortening of the cell cycle, and increase in symmetric cell division and a consequent increase in the number of adult-born interneurons and astrocytes in the olfactory bulb (OB). Using morphological analysis and electrophysiological recordings in acute OB slices, the authors then show that these supernumerous neurons mature and integrate into the OB network in a manner similar to adult-born OB neurons in wild-type mice. Finally, the authors use a go/no go odor discrimination task to assess the functional consequences of this genetic manipulation and report that the performance of 4D transgenic mice is improved compared to controls when discriminating between two similar odorant mixtures. The authors here characterize a new iteration of an elegant, previously established mouse model of 4D overexpression and report interesting effects of transient 4D overexpression on adult neurogenesis in the SVZ. However, a major claim of this study, improved odor discrimination with more adult-born OB interneurons, is only weakly supported by experimental data. In its present form, the manuscript is thus not suitable for publication in the EMBO Journal.

Major points:

- 1) The authors report that a moderate increase in adult neurogenesis in the SVZ, caused by transient 4D overexpression, improves odor discrimination. This improvement is only observed for the 'difficult' odor discrimination task, in which mice discriminate between two similar mixtures of (+) and (-) octanol, at low odor concentration. The analysis of this phenotype, a key point of the paper, should be corroborated in several ways. The authors should provide a more detail description of the experiments presented in Figure 4, including the performance of mice during all the octanol trials and the number of mice used in individual experiments. What was scored? The authors mention that a correct response is scored when mice 'wait and lick' - please specify. For the octanol mixture discrimination at 1% odor concentration, 4D transgenic mice appear to start the task at a performance level significantly above chance and similar to the performance level observed in the same task at low concentration - please clarify. More generally, the observation that enhanced accuracy of odor discrimination is only observed at a low odor concentration suggests that in 4D transgenic mice, odor detection threshold rather than discriminability may be altered. This possibility should be tested more systematically, using odors at a range of different concentrations. Finally, odor detection and discrimination tasks should be performed with more than one single odor pair.
- 2) In their histological and behavioral analyses, the authors compare nestin-creER^{+/+}, ROSA26-rtTA flox^{+/+}, tet4D-RFP^{+/+} transgenic mice with nestin-creER^{+/+}, ROSA26-rtTA flox^{+/+}, tet4D-RFP^{-/-} controls. The authors should clarify if littermates were compared in individual experiments, and the breeding strategy to generate mice of the various genotypes. How did the authors control for potential effects caused by the tet4D-RFP allele?
- 3) The slice electrophysiology experiments appear to lack quantification and statistical analyses. In addition to the example traces in Figure 3, individual data points, averages and statistics for key physiological parameters should be shown. The images of patched cells (Figure 3E) are of insufficient quality and uninformative.

Minor points:

- 4) A nuclear counterstain should be included in the histological analysis in Figure 2E as a reference

for the different layers of the OB.

5) The analysis of the neurogenic effects of 4D overexpression in Figures 1 and 2 would be more straightforward and convincing with the inclusion of the RCE-GFP allele used in Figure 3.

6) Statements about RFP stability on page 6 (referring to Figure S1A') and p8 (Figure 2D) appear contradictory - specify if RNA or protein was analyzed in main test and Figure legends.

7) English should be improved for better readability. The manuscript contains several typos/grammar mistakes and convoluted sentences.

Referee #3:

The authors present an interesting, elegantly designed and well characterised mouse model of transiently increased adult neurogenesis, and use it to ask the important question of whether the absolute number of newborn neurons in the OB can influence sensory function. Unfortunately, however, while the early portions of the study describing the process of elevated neurogenesis are on the whole well performed, the descriptions of the extra newborn OB granule cells as 'normal', and of olfactory discrimination as 'improved', are currently far from convincing.

Major points:

- 1) The 'highly significant improvement in olfactory performance' (abstract) is not convincing.
 - Firstly, the analysis of the data in Fig4C is inappropriate for a 1:10 dilution task carried out on animals already trained on the undiluted mixtures. There is no evidence for additional learning here, so essentially this entire experiment represents an extended probe trial. With no change in performance over time, it is entirely inappropriate to run a 2-way anova with trial block as a variable - such an analysis assumes there are 10 independent comparisons of 4D+/- performance on the task, and will erroneously increase the chance of a significant difference between the groups. It is also highly inappropriate to use unidirectional error bars on these plots. An appropriate analysis here should, at least, compare the 4D+/- groups based on purely the mean % correct performance in the first 100 trials of the diluted task (assuming both groups finished the previous undiluted task at equal performance levels). Or, if the authors want to use all of the 1000-trial data, take averages per mouse across all of those 1000 trials. Alternatively, perform a repeated-measures ANOVA, or any similar analysis, that takes into account the fact that the same mice are performing the task at all the different time points. Even better, though, would be to re-run the task for animals that had never learnt the undiluted version (though this would still need a repeated-measures ANOVA analysis). And please use bidirectional error bars!
 - For all behavioural tasks performed on a novel, custom-built olfactometer, the authors need to rule out alternative, non-olfactory explanations for task performance (and any potential group differences). Olfactometer tasks can notoriously be performed without using the olfactory cues, by animals detecting e.g. differences in auditory cues produced by the valve movements associated with different stimuli. If the mice are, at least partly, solving the tasks using non-olfactory cues, this could potentially account for the residual above-chance performance on the 1:10 dilution task. Alternatively, there may be larger differences in olfactory behaviour between the 4D+/- groups that are being masked by their use of alternative, non-olfactory strategies for task completion. These possibilities need to be addressed - for example, can the mice perform the discrimination tasks just as well if on random test trials the S+/- stimuli are just pure air? Or if the valve combinations are switched? Also, are there differences between groups in non-olfactory factors such as weight loss/dehydration, general movement levels, licking frequency/persistence etc etc that could possibly account for the (small) difference in slightly-above-chance 4D+/- performance on the diluted test?
 - Even if the 4D+/- difference survives more appropriate analysis, it is also an isolated effect on a one-off specific task. To be more confident of a reliable effect of elevated adult neurogenesis on discrimination performance, it would be best to replicate the 4D+/- difference with an alternative 'very tough' task (with naïve animals, and R-M analysis), e.g. a (diluted?) 45/55 mixture, ideally with different odorants.
 - Why do the DTs not scale as expected with task difficulty? In particular, if the 1:10 dilution task is indeed the most difficult, why is it 'solved' more quickly than the 40/60% discrimination task at full

concentration?

2) The evidence for supernumerary neurons being 'undistinguishable from physiologically generated ones at the molecular, morphological and electrophysiological level' is fragmentary and unconvincing:

- For the analyses in Fig3, it is unclear whether GFP+/RFP- and GFP+/RFP+ cells are compared within the same mice, or between different mice. Making both comparisons would be useful - within-mouse to directly compare 4D+/- neurons generated at the same time from the same niche, and between-mouse to address the potential for an overall effect of elevated GC number on all newborn GCs. Though understandably more laborious, a more appropriate negative control would have been a triple transgenic mouse with the genotype: nestin-CreERT2, ROSA26-rtTA-flox, tet-RFP (same construct as the one used for 4D mice, but lacking the Cdk4 and Cyclin D1 coding regions). These mice would show RFP expression only on the newborn neurons generated after tam and dox administration, thus allowing for a better comparison with the supernumerary RFP+ neurons of 4D+ mice.
- If the 4D+ neurons still retain RFP 30 days after dox washout, could they presumably also still have elevated levels of cdk4/cyclinD1? They would therefore not be 'molecularly normal' (indeed, where are the data for comparisons at the 'molecular' level mentioned in the abstract?). This would not be such a problem for interpretations of any behavioural effects if the morphological/physiological description of these cells as 'normal' were convincing, but unfortunately this is not the case (see below).
- 'Morphological' analysis is restricted to counts of spine density. Especially given that the authors already have tissue containing sparsely-labelled cells, GC dendritic morphology should also be fully reconstructed and quantitatively compared (e.g. total branch length, Sholl analysis, location in GCL etc etc)
- The immuno/EM synaptic characterisation in Fig3c/d is disappointingly brief and superficial - this is a real shame, especially given the great effort involved in generating EM immunogold images. A quantitative comparison is needed of RFP+/- cells, at least of the densities of pre/post-synaptic specialisations in both preparations, and ideally of other parameters (e.g. PSD size, vesicle number, identity of pre & post-synaptic partners etc etc)
- The electrophysiological recordings presented in Fig3e do not currently come from enough cells to be able to make strong statements about (a lack of) group differences, and they lack descriptions of important parameters such as membrane capacitance, voltage threshold, rheobase, spike width, and input-output function. An example image should also be included to show how RFP+ cells were targeted for recordings.

Minor points:

- Fig1 shows an increased % of BrdU+ cells amongst RFP+ neurons in 4D+ mice (c), and then shows an increase in overall cell numbers in the SVZ of these mice (d). What's missing is the intermediate characterisation (as far EdU in Fig2): what's the % of BrdU+ cells of each type, regardless of RFP expression?
- Could the transient increase in newborn cells produced in 4D+ mice be immediately followed by a compensatory drop in neurogenesis? This could be tested by, e.g. assessing numbers of BrdU+ cells 30d after BrdU pulses applied immediately after the termination of Dox treatment. If this were the case, the total number of newborn neurons generated in the week starting with the dox treatment might actually be no different in 4D+ and 4D- mice, and this might explain the lack of clear behavioural differences between the groups.
- Statistical tests:
 - a) The authors have used multiple individual t-tests to compare groups within a given experiment (e.g. Fig1C/D, Fig2E). This approach can generate spurious false positive results in terms of significance. In these cases, and potentially elsewhere in the paper, the authors should perform 2-way ANOVA analyses (which will probably reveal a simple overall effect of 4D+ across all cell types).
 - b) Statistical tests for differences in proportions (e.g. Fisher's Exact Test) should be performed for the data in Fig2C/E
 - c) Everywhere, much more comprehensive reporting is needed of all quantitative comparisons: for every test, please report the mean+SEM(or SD) values for each group, test statistic (e.g. t value), df, and actual p value (not just $p < 0.05$)
- Methodological detail:
 - a) For image analysis, how exactly how were cells determined to be positive for individual markers?

Were experimenters always blind to group here? Was co-label evaluated in 3d?

b) The electrophysiology section lacks crucial information on individual protocols, sampling frequencies, quality control (series resistance, capacitance compensation, bridge balance), and how measures were generated (e.g. how was V_m determined? And R_{in} ?).

1st Revision - authors' response

28th Sep 2018

Please see next page

AUTHORS' GENERAL STATEMENT

We thank the reviewers for consistently finding our work interesting, elegant and important and for the many constructive points of criticism that helped us to considerably strengthen our work. We believe that all the reviewers' questions and requests are now satisfactorily addressed in a thoroughly revised manuscript (red text) and point-by-point responses below. Our reply below also includes 9 new figures from further experiments and unpublished data. For simplicity and brevity, these figures were not included in the main manuscript but would become available in case of publication and based on EMBO's transparent review process.

REFEREE #1:

The authors examine the effects of induced neural stem cell expansion in the SVZ on OB neurogenesis and function. The experiments are well executed and the findings are consistent with previous studies (using optical stimulation and LOF approaches) implicating OB neurogenesis in discrimination of similar odor stimuli. The authors provide compelling evidence for a powerful way to amplify the population of neural stem cells in vivo to enhance neurogenesis. I think it would strengthen the study if the authors can provide some evidence for changes in downstream circuits like the Piriform cortex in these mice. One way to assess this is by c-Fos immunohistochemistry in 4D+ and 4D- mice following discrimination.

Authors' response

We thank the reviewer for finding our study well executed and compelling. We have performed the suggested experiment and quantified c-Fos in layer II of the anterior and posterior piriform cortex of brains preserved after the olfactometry tests shown in Fig. 4D. This revealed an intriguing increase by ca. 40% in 4D+ mice (from 251 ± 76 to 346 ± 166 cells/mm³). Yet, this increase was not significant (n=3; p<0.4; **Reviewer Fig. 1**). In this context, studies have shown that different odorants (or different concentrations of the same odorant) trigger differential c-Fos activity in the piriform cortex (among others: Illig et al., J Comp Neurol, 2003; Stettler et al., Neuron, 2009; Apicella et al., J Neurosci, 2011). Yet, to our knowledge it is not known whether the same odorant and concentration would change c-Fos in mice whose only difference is olfactory performance. Given this lack of knowledge and the absence of a simple formalism that would predict the outcome of this experiment, we feel that our observed lack of significant change cannot be interpreted and for this reason we included these data only in this reply to the reviewer but not in the revised manuscript.

Reviewer Fig. 1. C-Fos quantification in the piriform cortex of brains collected after the olfactometry tests show in Fig. 4D. See text above for details. Scale bar=100µm; error bars=SD.

REFeree #2:

In this manuscript, Bragado Alonso and colleagues use mouse genetics to selectively target Cdk4/cyclinD1 (4D) overexpression to neural stem cells of the subventricular zone (SVZ). The authors find that transient 4D overexpression leads to a shortening of the cell cycle, and increase in symmetric cell division and a consequent increase in the number of adult-born interneurons and astrocytes in the olfactory bulb (OB). Using morphological analysis and electrophysiological recordings in acute OB slices, the authors then show that these supernumerous neurons mature and integrate into the OB network in a manner similar to adult-born OB neurons in wild-type mice. Finally, the authors use a go/no go odor discrimination task to assess the functional consequences of this genetic manipulation and report that the performance of 4D transgenic mice is improved compared to controls when discriminating between two similar odorant mixtures.

The authors here characterize a new iteration of an elegant, previously established mouse model of 4D overexpression and report interesting effects of transient 4D overexpression on adult neurogenesis in the SVZ. However, a major claim of this study, improved odor discrimination with more adult-born OB interneurons, is only weakly supported by experimental data. In its present form, the manuscript is thus not suitable for publication in the EMBO Journal.

Authors' response

We thank the reviewer for the interest in our work and constructive points of criticisms that, we believe, are now addressed in the revised manuscript and point-by-point responses below.

Major points:

Referee #2; point 1

The authors report that a moderate increase in adult neurogenesis in the SVZ, caused by transient 4D overexpression, improves odor discrimination. This improvement is only observed for the 'difficult' odor discrimination task, in which mice discriminate between two similar mixtures of (+) and (-) octanol, at low odor concentration. The analysis of this phenotype, a key point of the paper, should be corroborated in several ways. The authors should provide a more detail description of the experiments presented in Fig. 4, including the performance of mice during all the octanol trials and the number of mice used in individual experiments. What was scored? The authors mention that a correct response is scored when mice 'wait and lick' - please specify. For the octanol mixture discrimination at 1% odor concentration, 4D transgenic mice appear to start the task at a performance level significantly above chance and similar to the performance level observed in the same task at low concentration - please clarify.

More generally, the observation that enhanced accuracy of odor discrimination is only observed at a low odor concentration suggests that in 4D transgenic mice, odor detection threshold rather than discriminability may be altered. This possibility should be tested more systematically, using odors at a range of different concentrations. Finally, odor detection and discrimination tasks should be performed with more than one single odor pair.

Authors response

We regret our lack of clarity and try now to address all the points raised by this reviewer breaking them down in five consecutive sections as follows:

i) **Better description of protocols:** We have now extended the text, materials and methods, figure legends and supplemental information (SI) including mice number, parameters assessed, methods, etc. (see in particular red text in Methods page 19-22 and throughout SI).

ii) **Performance during all octanol trials:** All mice did at least 1,000 trials but differed in the total number of trials performed by the end of the testing with some performing more than 1,000 trials. To account for this and normalize this variable, we always compared all the first 1,000 trials among the different genotypes, as shown in Fig. 4 and S4 and described in SI. This clearly does not apply to the probe trial in which only the first 200 trials were tested as requested by Reviewer 3, point 1, section *ii* & *iii* (see page 8 of this rebuttal).

iii) **Mice appear to start the tasks at a performance above chance:** It should be noted that in “easy” tasks with concentrated odorants (e.g. Fig. 4C and S4B) learning is rather fast starting already within the first few trials. Hence, performances of 50% are usually limited to the first 10-20 trials. When binning 100 trials together to assess percentages these 10-20 trials have a lesser impact on the final percentage calculated including the following 80-90 trials and characterized by performances above 50%. This effect is even stronger with mice familiar with the test, which further speeds up their learning. Furthermore, when the same odorants are used at a higher dilution (e.g. Fig. 4D), mice do not seem to react to the previously learnt odorants as new odors *per se* and the drop in performance in switching from concentrated to diluted ones is minor (Nunes et al., Nat Commun 2015; Shimshek et al., PLoS Biol, 2005; Gschwend et al., Nat Neurosc, 2015 and **Reviewer Fig. 2**). Together, this explains why mice appear to start at performances above chance.

Reviewer Fig. 2. Performance at decreasing dilutions of the same odorants (AA vs. EB) in wild type mice. In relation to section *iii* above, note that mice start subsequent tests at performances above 50%. Wild type (grey), GluA2 mutant mice (black) (Kudryavitskaya and Kuner, unpublished data).

iv) **Detection threshold vs. discriminability:** In principle the reviewer is correct that the detection threshold (sensitivity) could differ between genotypes. However, our unpublished data show that mice reach a detection threshold only at much higher dilutions ($<10^{-5}$ - 10^{-6}) than those used here (10^{-1}) (**Reviewer Fig. 2**). Since our dilution of octanols is $>10,000$ fold higher than this threshold, we conclude that increased neurogenesis improves discriminability. Also, while we could not find any information about the detection threshold for octanol in mice, values for humans are in the range of 110-130 parts per billion (roughly 1×10^{11} molecules) (Buttery et al., J Agric Food Chem, 1988; Baker, J Am Water Works Assoc,

1963). In our reservoirs for the 1:10 dilution, even the less concentrated octanol enantiomer (the 40% component) in the vapor phase is in the range of $2-5 \times 10^{12}$ molecules so still one order or magnitude higher than the threshold for humans. Although we know that these values cannot directly be translated to mice, combined with the data by Kudryavitskaya and previous experiments (**Reviewer Fig. 2** and not shown) we feel confident that mice can detect the 1:10 diluted odors. We have now clarified in the revised text that concentrations were far from the detection threshold, a new aspect of olfaction in 4D+ mice not addressed in our study and likely requiring further optimization of our approach (see Discussion, page 19).

v) **Tasks performed with more than one odor pair:** We address this by distinguishing the two contexts of “easy” vs. “difficult” tasks. For the former, different odor pairs (five) were indeed used conforming to the suggestion of the reviewer (**Reviewer Fig. 3**) and in which no difference was detected (Fig. 4C and S4B). From here, our next goal was to make the task as difficult as possible. Yet, not too difficult to allow mice to perform above random choice. This is exactly what we achieved with the 1:10 dilution of octanol mixtures with performances of ca. 70%, that is right between 50% of random choice and 95% of easy tasks.

TASK #	COHORT 1	COHORT 2
1	Cineole (1%) vs Eugenole (1%)	Amylacetate (1%) vs Ethylbutyrate (1%)
2	Amylacetate (1%) vs Ethylbutyrate (1%)	60/40 Amylacetate/Ethylbutyrate Mix vs 60/40 Ethylbutyrate/Amylacetate Mix
3	60/40 Amylacetate/Ethylbutyrate Mix vs 60/40 Ethylbutyrate/Amylacetate Mix	(+)-Octanole (1%) vs (-)-Octanole
4	(+)-Octanole (1%) vs (-)-Octanole	60/40 (+)-Octanole/(-)-Octanole Mix vs 60/40 (-)-Octanole/(+)-Octanole Mix
5	60/40 (+)-Octanole/(-)-Octanole Mix vs 60/40 (-)-Octanole/(+)-Octanole Mix	1: 10 diluted 60/40 (+)-Octanole/(-)-Octanole Mix vs 1: 10 diluted 60/40 (-)-Octanole/(+)-Octanole Mix

Reviewer Fig. 3. Odour pairs used in the two cohorts of mice in our study. Percentages in brackets denote the final concentration upon dilution in mineral oil.

Next, in the context of difficult odors, while agreeing with the principle that more pairs might yield new insight, we have deliberately chosen an odor pair that we know to belong to the most difficult monomolecular odorants for mice to discriminate. **Reviewer Fig. 4** shows the discrimination time (DT) as a function of Euclidean distance (<https://doi.org/10.1101/356279>). The blue symbols represent octanols as monomolecular (filled) and binary mixtures (open). The more similar the glomerular representation, the more difficult the task and longer the DT. (Note that DTs in **Reviewer Fig. 4** were determined at discrimination accuracies >90% and hence cannot be directly compared to the values for the 1:10 dilution at 70% discrimination.) For this reason, we are not aware of any other monomolecular odor pair that would provide us with a significantly more difficult task than the one already used. This is an important aspect, now discussed in our manuscript (page 15), and thank the reviewer for this point. (See also response to reviewer 3, point 1, section v.)

Reviewer Fig. 4. Discrimination time at 90% performance as a function of Euclidean distance for pure odorants (filled circles) or 40/60% binary mixtures (empty circles). Low Euclidean distance represents high similarity. Green: cineole/eugenole; red: amylacetate/ethylbutyrate; blue: (+)/(−)-octanol; purple: (+)/(−)-carvone (doi: <https://doi.org/10.1101/356279>). Note that the binary mixture of octanol enantiomers (blue arrow) was additionally diluted 1:10 in Fig. 4D indicating the extreme difficulty of this task.

We hope that the sections *i-v* above, each now briefly discussed in our manuscript, satisfactorily address the issues raised by this reviewer about clarity, performances above 50%, discriminability vs detection threshold and using more difficult odorant pairs.

Referee #2; point 2

In their histological and behavioral analyses, the authors compare nestin-creER^{+/+}, ROSA26-rtTA flox^{+/+}, tet4D-RFP^{+/+} transgenic mice with nestin-creER^{+/+}, ROSA26-rtTA flox^{+/+}, tet4D-RFP^{-/-} controls. The authors should clarify if littermates were compared in individual experiments, and the breeding strategy to generate mice of the various genotypes. How did the authors control for potential effects caused by the tet4D-RFP allele?

Authors' response

The triple line was generated from the three heterozygous lines previously described (Imayoshi et al., *Nat Neurosci*, 2008; Belteki et al., *Nucleic Acids Res*, 2005; Nonaka-Kinoshita et al., *EMBO J*, 2013). The genetic background was homogenized after obtaining nestin-cre^{+/+}, R26-rtTA^{+/+}, tet-4D^{+/-} mice (only the latter allele as heterozygous) for >5 generations and finally bred to triple homozygous (4D^{+/+} and 4D^{-/-}). All experiments used mice from different litters and same genetic background. The only exception was the 2 month-long BrdU labeling in Fig. S2, which additionally also included littermates generated by crossing 4D^{+/-} mice with no difference in phenotype observed relative to other experiments. Finally, cross with RCE-GFP (Miyoshi et al., *J Neurosci*, 2010), also homozygous, led to quadruple heterozygous mice, which was necessary since GFP and rtTA are within the same R26 locus. Also here, no difference in phenotype, cell numbers, BrdU, etc was observed between quadruple-hetero 4D^{+/-} and triple-homo 4D^{+/+} as described in the revised SI (page 2).

With regard to potential side-effects of the 4D allele, we controlled this as requested by BrdU administration in 4D^{-/-} and 4D^{+/+} mice but without tamoxifen nor doxycycline. In essence, proliferation in the SVZ was assessed in the presence of the two 4D alleles but without inducing 4D expression. Numbers of BrdU⁺ cells after 12 h labeling were virtually identical in 4D^{-/-} and 4D^{+/+}, ($570 \pm 93 \times 10^3$ and $539 \pm 60 \times 10^3$ per mm³ respectively; $p < 0.7$ unpaired t-test, $n = 3$; **Reviewer Fig. 5**) excluding side-effects by insertion of the 4D alleles.

Reviewer Fig. 5. BrdU incorporation in the SVZ of 4D^{-/-} and 4D^{+/+} mice but without administration of neither tamoxifen nor doxycycline to test for the genomic effect of 4D insertion without 4D expression. See above for details. Bar=50μm; error bars=SD

Referee #2; point 3

The slice electrophysiology experiments appear to lack quantification and statistical analyses. In addition to the example traces in Fig. 3, individual data points, averages and statistics for key physiological parameters should be shown. The images of patched cells (Fig. 3E) are of insufficient quality and uninformative.

Authors' response

The reviewer is correct and we have extensively expanded this section both with regard to the number of cells for statistics as well as parameters measured per cell. The latter now includes number, width and amplitude of spikes, membrane resting potential, input resistance and 6 additional parameters included in the new Fig. 3 and S3 that very comprehensively describe the activity of neurons derived from 4D-manipulated NSC. Confirming our conclusions, no significant difference was observed in any parameter supporting the notion that 4D-derived neurons are physiologically normal. We agree with the reviewer that the images of patched cells were low quality and uninformative and for this reason we have removed them.

Minor points

Referee #2; point 4

A nuclear counterstain should be included in the histological analysis in Fig. 2E as a reference for the different layers of the OB.

Authors' response

Fig. 2E and S2C have been modified accordingly as requested.

Referee #2; point 5

The analysis of the neurogenic effects of 4D overexpression in Figs. 1 and 2 would be more straightforward and convincing with the inclusion of the RCE-GFP used in Fig. 3.

Authors' response

Unfortunately this was not possible due to limitations in fluorescence channels and the need to co-stain multiple markers (2+DAPI) hence limiting the choice of the 4th label as either RFP or GFP, but not both. Of the two, RFP is superior since this is linked to 4D by 2A-cleaving peptides providing a direct read out of 4D expression.

Referee #2; point 6

Statements about RFP stability on page 6 (referring to Fig. S1A') and p8 (Fig. 2D) appear contradictory-specify if RNA or protein was analyzed in main test and Fig. legends.

Authors' response

We revised the text and figure legends accordingly throughout the text.

Referee #2; point 7

English should be improved for better readability. The manuscript contains several typos/grammar mistakes and convoluted sentences.

Authors' response

We now revised the entire manuscript and hope this to be to the reviewer's satisfaction.

REFEREE #3:

The authors present an interesting, elegantly designed and well characterised mouse model of transiently increased adult neurogenesis, and use it to ask the important question of whether the absolute number of newborn neurons in the OB can influence sensory function. Unfortunately, however, while the early portions of the study describing the process of elevated neurogenesis are on the whole well performed, the descriptions of the extra newborn OB granule cells as 'normal', and of olfactory discrimination as 'improved', are currently far from convincing.

Authors response

We thank the reviewer for finding our study interesting and elegant and hope to have addressed his/her concerns in the revised manuscript and point-by-point responses below.

Major points:**Referee #3, point 1**

- i) The 'highly significant improvement in olfactory performance' (abstract) is not convincing.
- ii) Firstly, the analysis of the data in Fig.4C is inappropriate for a 1:10 dilution task carried out on animals already trained on the undiluted mixtures. There is no evidence for additional learning here, so essentially this entire experiment represents an extended probe trial. With no change in performance over time, it is entirely inappropriate to run a 2-way anova with trial block as a variable - such an analysis assumes there are 10 independent comparisons of 4D+/- performance on the task, and will erroneously increase the chance of a significant difference between the groups.
- iii) It is also highly inappropriate to use unidirectional error bars on these plots. An appropriate analysis here should, at least, compare the 4D+/- groups based on purely the

mean % correct performance in the first 100 trials of the diluted task (assuming both groups finished the previous undiluted task at equal performance levels). Or, if the authors want to use all of the 1000-trial data, take averages per mouse across all of those 1000 trials. Alternatively, perform a repeated-measures ANOVA, or any similar analysis, that takes into account the fact that the same mice are performing the task at all the different time points. Even better, though, would be to re-run the task for animals that had never learnt the undiluted version (though this would still need a repeated-measures ANOVA analysis). And please use bidirectional error bars!

iv) For all behavioural tasks performed on a novel, custom-built olfactometer, the authors need to rule out alternative, non-olfactory explanations for task performance (and any potential group differences). Olfactometer tasks can notoriously be performed without using the olfactory cues, by animals detecting e.g. differences in auditory cues produced by the valve movements associated with different stimuli. If the mice are, at least partly, solving the tasks using non-olfactory cues, this could potentially account for the residual above-chance performance on the 1:10 dilution task. Alternatively, there may be larger differences in olfactory behaviour between the 4D+/- groups that are being masked by their use of alternative, non-olfactory strategies for task completion. These possibilities need to be addressed - for example, can the mice perform the discrimination tasks just as well if on random test trials the S+/- stimuli are just pure air? Or if the valve combinations are switched? Also, are there differences between groups in non-olfactory factors such as weight loss/dehydration, general movement levels, licking frequency/persistence etc etc that could possibly account for the (small) difference in slightly-above-chance 4D+/- performance on the diluted test?

v) Even if the 4D+/- difference survives more appropriate analysis, it is also an isolated effect on a one-off specific task. To be more confident of a reliable effect of elevated adult neurogenesis on discrimination performance, it would be best to replicate the 4D+/- difference with an alternative 'very tough' task (with naïve animals, and R-M analysis), e.g. a (diluted?) 45/55 mixture, ideally with different odorants.

vi) Why do the DTs not scale as expected with task difficulty? In particular, if the 1:10 dilution task is indeed the most difficult, why is it 'solved' more quickly than the 40/60% discrimination task at full concentration?

Authors response

i) *The improvement in olfactory performance is not convincing:* We appreciate that the effect size, or magnitude, of the improvement may appear relatively small with a change from ca. 65% to only 72% (Fig. 4D). However, since a performance of 50% is expected from random choice, this is equivalent to say that relative to controls 4D+ mice have improved by half from 15% to 22% above random choice. Whether or not this is a small effect is arguable. Remains the fact that this improvement was statistically significant and biologically relevant given the importance of olfaction for mice (see *ii* & *iii* below for the assessments of significance). Nonetheless, we now tried to more objectively discuss the implications of comparing magnitude of the effect vs. its significance while toning down some of our claims in the abstract, main text and discussion (red text throughout the manuscript).

ii & iii) **Evidence for additional learning and use of proper statistic:** Concerning the diluted task, the reviewer is correct that this is more properly defined as a probe trial after learning the concentrated odors in which mice indeed have finished the test at the same performance (Fig. 4C). Hence, the reviewer is correct in considering t-test more appropriate. We have corrected this as requested by Student's t-test considering unequal variance and compared the performances of 4D- and 4D+ mice (n=15 and 10, respectively) in the first 2 bins (200 trials). This confirmed the significant increase in performance of 4D+ mice ($p<0.03$) (new Fig. 4D). Furthermore, we additionally corroborated this effect by assessing a change in licking frequency which is defined as the proportion of trials in which mice licked (irrespective of the odour being presented). While 4D+ mice maintained a ca. 50% lick frequency during the diluted octanols trials, as expected by the fact that 50% of the trials are S+, 4D- displayed an increase in frequency to 65%. In essence, the lower performance of controls was concomitant with them licking more often irrespective of the odor presented to compensate for the average decrease in water obtained relative to 4D+. Also this effect was statistically significant ($p<0.01$) as assessed by Student's t-test (new Fig. 4D).

iv) **Non-olfactory explanations for task performance:** Also here, the reviewer is entirely correct and a comprehensive list of tests was indeed performed to exclude non-olfactory cues that were not described in our original manuscript for brevity. These included testing the performance of the same cohort of animals first without, and subsequently with, olfactory cues. When only mineral oil was used, mice could not discriminate S+ vs S- trials even over thousands of attempts. This, together with other parameters (**Reviewer Fig. 6A-E**), validated the use of the olfactometer to specifically detect olfactory cues.

Reviewer Fig. 6. Discrimination of mineral oil vs mineral oil. **A)** Performances showing no increase over chance over thousands of trials. **B)** Sensitivity index (d -Prime) as a measure of how animals separate two signals confirming lack of discrimination. **C-D)** Indicators of motivation and arousal for discrimination as in A showing Inter-Trial Intervals with no clear change over thousands of trials (C), decrease in sampling frequency after ca. 1,000 trials reflecting decreased motivation (D) and lick frequency remaining excessively high reflecting a lack of cues needed to distinguish S+ vs S- trials. n=3, adult male C57BL/6 wt animals). Data as mean \pm SEM; Reinert and Kuner, unpublished.

The same animals were subsequently trained on the same olfactometer but this time using odorants and yielding essentially the same degree of learning and performance as already described in our study (Fig. 4C and S4B). Hence, mice that were unable to learn a mineral oil only task were able to discriminate odor cues. Also important in this context, we presented the odorants from different valves while randomly switching the reservoirs to different valves over the course of the testing to ensure that mice could not use other cues such as the sound of individual valves or marginal differences in odour onset depending on the valve used. This is now explained in our revised SI section (Page 7-8).

Furthermore, a number of parameters were also compared in 4D- and 4D+ mice prior to, during, and after the olfactometry tests that were not mentioned in our manuscript. These included body weight, circadian activity, lick frequency and others, which did not significantly change within the course of the experiment. As one specific example, while individual mice had different periods of the day when they preferred to perform their trials, no correlation was found with their discrimination performance nor was any significant difference found between 4D- vs 4D+ mice (**Reviewer Fig. 7**). A similar lack of correlation, nor difference, resulted when comparing body weight over the course of the tests, lick frequency, inter-trial time, sampling frequency, and many other parameters that we assessed but were not mentioned in the original manuscript for brevity and simplicity.

Reviewer Fig. 7. Circadian rhythm of mice represented as the percentage of trials (radius) during a day (clock) of diluted octanols testing (line=mean; shaded area=SEM; differences not significant). Light 6am-6pm.

v) **Replicate with alternative “very tough” tasks:** As pointed out in response to Reviewer 2, point 1, section v, we already tested five pairs of odorants plus one “very tough” 40/60% mixture of octanol enantiomers that, to the best of our knowledge, are among the most difficult monomolecular odorants available for mice to discriminate (**Reviewer Fig. 3 and 4**). While in principle we agree that equally difficult, or if possible even more difficult, new tasks might provide additional insight, we believe that the conditions already tested are sufficient to corroborate the central claim of our study. Moreover, as now discussed in detail in the revised text (page 19), it is very likely that to assess such new aspects on olfaction with more difficult tasks we would first need to optimize our 4D model beyond its current limits of induction in ca 50% of NSC and/or increasing neurons beyond the current 5% of the total. Altogether, we seriously doubt that exploring these new aspects may - at the current time and current limitations - justify the additional work and time that these experiments require including the approval of a new animal license for the sacrifice of mice (ca 5-6 months) and additional time for inducing of the 4D model (2 months) and subsequent behavioural training

(3-4 months). Therefore, in response to this comment, we revised our Discussion to emphasize that a more extensive coverage of odor stimuli might in the future reveal additional effects of neurogenesis that were not assessed in this study (page 19).

vi) *Why is the 1:10 dilution task solved more quickly than at full concentration?* We respectfully disagree with the statement that the discrimination of the 1:10 mixture is faster than at full concentration. First, these differences were not significant ($p < 0.1$ and $p < 0.5$ for 4D- and 4D+, respectively; **Reviewer Fig. 8**). Second, it is technically not correct to compare these discrimination times since mice showed different performances in each task (95% and 70%) thus different thresholds were used to calculate these DTs. Third, on a more theoretical ground, an apparent low discrimination time in a very tough task could be influenced by mechanisms adapting to a new speed/accuracy-tradeoff. For example, when mice notice that a high performance (i.e. likelihood of reward) cannot be achieved, they revert to strategies favoring a fast decision because this leads to more rewards (i.e. water per unit of time) (Zacksenhouse et al 2010, Ratcliff 1978, Holmes & Cohen 2014). For these reasons we agree that the DTs of the diluted octanols were misleading and have removed them from the revised manuscript.

Reviewer Fig. 8. Discrimination time across the different tasks. Comparisons among the same genotype done by paired t-test and among different genotype by unpaired t-test. Note that the performance threshold used to calculate DT was 95% (left) and 70% (right).

Referee #3, point 2

The evidence for supernumerary neurons being 'undistinguishable from physiologically generated ones at the molecular, morphological and electrophysiological level' is fragmentary and unconvincing:

i) For the analyses in Fig. 3, it is unclear whether GFP+/RFP- and GFP+/RFP+ cells are compared within the same mice, or between different mice. Making both comparisons would be useful - within-mouse to directly compare 4D+/- neurons generated at the same time from the same niche, and between-mouse to address the potential for an overall effect of elevated GC number on all newborn GCs. Though understandably more laborious, a more appropriate negative control would have been a triple transgenic mouse with the genotype: nestin-CreERT2, ROSA26-rtTA-flox, tet-RFP (same construct as the one used for 4D mice, but lacking the Cdk4 and Cyclin D1 coding regions). These mice would show RFP expression only on the newborn neurons generated after tam and dox administration, thus allowing for a better comparison with the supernumerary RFP+ neurons of 4D+ mice.

ii) If the 4D+ neurons still retain RFP 30 days after dox washout, could they presumably also still have elevated levels of cdk4/cyclinD1? They would therefore not be 'molecularly normal' (indeed, where are the data for comparisons at the 'molecular' level mentioned in the abstract?). This would not be such a problem for interpretations of any behavioural effects if the morphological/physiological description of these cells as 'normal' were convincing, but unfortunately this is not the case (see below).

iii) 'Morphological' analysis is restricted to counts of spine density. Especially given that the authors already have tissue containing sparsely-labelled cells, GC dendritic morphology should also be fully reconstructed and quantitatively compared (e.g. total branch length, Sholl analysis, location in GCL etc etc)

iv) The immuno/EM synaptic characterisation in Fig.3c/d is disappointingly brief and superficial - this is a real shame, especially given the great effort involved in generating EM immunogold images. A quantitative comparison is needed of RFP+/- cells, at least of the densities of pre/post-synaptic specialisations in both preparations, and ideally of other parameters (e.g. PSD size, vesicle number, identity of pre & post-synaptic partners etc etc).

v) The electrophysiological recordings presented in Fig.3e do not currently come from enough cells to be able to make strong statements about (a lack of) group differences, and they lack descriptions of important parameters such as membrane capacitance, voltage threshold, rheobase, spike width, and input-output function. An example image should also be included to show how RFP+ cells were targeted for recordings.

Authors response

i) *Comparison within and across mice and generation of a tet-RFP line:* 4D- and 4D+ cells were indeed compared both within as well as between mice and phenotypes consistent in both conditions. With regard to control tet-RFP mice, these could not be obtained since the lines were generated by pronuclear injection resulting in random genome integration. We do agree that such a control line would have been important, specifically if either of two conditions would apply: a) if 4D expression had a bias toward a specific sub-population of cells and/or b) to control for potential mutations caused by the insertion of the 4D cassette irrespective of its expression. The first possibility was excluded already by analyses of acute induction showing no bias in 4D expression (Fig. 1B and S1C). The second possibility is now also excluded by new experiments in which a similar abundance and proliferative activity of neural progenitors was found in 4D+ mice without inducing 4D expression (assessed by BrdU incorporation in 4D+ mice without tamoxifen/doxycyclin administration; see also Reviewer 2, point 2 and **Reviewer Fig. 5**). Since both possibilities were directly excluded, we conclude that a tet-RFP control line was not necessary.

ii) *Can neurons retain 4D and where are the data for comparisons at the 'molecular' level?* 4D retention is an important point addressed in previous studies. Briefly, cyclins are degraded at the end of each cell cycle phase, G1 for cyclinD1, hence their name: "cyclins". Since 4D is only induced in NSC (Fig. 1 and S1) neurons cannot inherit cyclinD1 that in contrast to RFP was degraded in the G1 of their mother cells. We have shown in the past that this also applies

to ectopically expressed cyclins (cyclinD1 and E). When transiently induced in NSC, these were undetectable in neurons resulting from them (Lange et al., *Cell Stem Cells*, 2009; Nonaka-Kinoshita, *EMBO J*, 2013). With regard to Cdk4, while possibly inherited by the neurons, this cannot have any activity in the absence of cyclinD1, hence its name: “cyclin-dependent kinase”. Moreover, we also previously showed that even when 4D is ectopically induced in newborn and mature neurons rather than stem cells (which is not the case in this study!), this does not trigger any obvious effect on neuronal maturation, migration, specification, etc neither during development nor adulthood (Nonaka-Kinoshita, *EMBO J*, 2013; Artegiani et al., *J Exp Med*. 2011; Berdugo-Vega et al., unpublished). All this is now discussed in the revised manuscript (page 17-18).

The reviewer also asks: “*where are the comparisons at the 'molecular' level?*”. Admittedly, we did not perform transcriptomics but Figs. 1, 2 and S2 show quantifications of several molecular markers including Dcx, NeuN, CalB, CalR and TH. Certainly, more could be added to this list but we believe that these well established markers are sufficient to identify newborn and mature neurons at the molecular level. Retention of 4D, and the possibility of subtle differences that went undetected, are now discussed in the revised text.

iii) ***Additional morphological analyses:*** We thank the reviewer for this suggestions and added to our study Sholl analyses and total branch length again revealing no detectable difference between physiologically and 4D-generated neurons (new Fig. 3B).

iv) ***EM quantification of pre/post-synaptic specialisations:*** In principle the reviewer is correct and we would love to be able to do EM quantifications. Unfortunately, however, the high quality of membrane contrast that is needed to perform these analyses is incompatible with the immunogold labeling protocols essential to identify RFP+ cells. Immunogold labeling uses permeabilization steps, detergents, silver enhancement, etc that notoriously lead to poor membrane contrast hampering a quantitative assessment of pre/post-synaptic specialisations. Under these conditions, finding individual RFP+ synapses with a “barely decent” membrane contrast on 100 nm EM sections is equivalent to finding needles in a haystack whose statistical assessment would require an amount of time/persons well beyond our resources. Not least, a normal pre/post-synaptic specialisation can be inferred from the normal electrophysiological activity of the neurons recorded, whose analysis now includes a dozen of parameters (see next point). The reviewer and readers should consider our EM analysis at the qualitative, rather than quantitative, level as one more evidence, in addition to many others, that 4D-derived neurons can physiologically mature and integrate as expected by the fact that no functional transgene is induced in these neurons (see section *ii* above).

v) ***Electrophysiological recordings and number of cells:*** We agree and have thoroughly expanded this section including more cells (12 4D– and 9 4D+) and comparing them both within the same section and across mice. Statistical analyses were made for the parameters previously measured as well as the new ones suggested by this reviewer (e.g. capacitance, voltage threshold, rheobase and others). In a total of 12 sets of parameters, neurons of 4D– and 4D+ mice were, de facto, statistically indistinguishable (Fig. 3E-H and S3B).

Referee #3, Minor points:

1) Fig.1 shows an increased % of BrdU+ cells amongst RFP+ neurons in 4D+ mice (c), and then shows an increase in overall cell numbers in the SVZ of these mice (d). What's missing is the intermediate characterisation (as far EdU in Fig.2): what's the % of BrdU+ cells of each type, regardless of RFP expression?

Authors response

1) We performed the requested calculation of BrdU+ cells in 4D mice irrespective of RFP expression. As expected, values fit well between 4D- controls and 4D+/RFP+ cells (**Reviewer Fig. 9**). Since in 4D+ mice only a fraction of neural progenitor cells are RFP+ (Fig. 1B), it logically follows that these values represent an average between the two.

Reviewer Fig. 9. BrdU quantification across the SVZ of mice treated as in Fig. 1. Markers for B, C, A cells and neurons are indicated (from left to right, respectively). n=3, unpaired t-test, *p<0,05, **p<0,01.

Referee #3, Minor points:

2) Could the transient increase in newborn cells produced in 4D+ mice be immediately followed by a compensatory drop in neurogenesis? This could be tested by, e.g. assessing numbers of BrdU+ cells 30d after BrdU pulses applied immediately after the termination of Dox treatment. If this were the case, the total number of newborn neurons generated in the week starting with the dox treatment might actually be no different in 4D+ and 4D- mice, and this might explain the lack of clear behavioural differences between the groups.

Authors response

This was an interesting possibility and we performed the experiment suggested confirming the increase in newly generated neurons in 4D+ mice even 30 days after the transitory induction of 4D ($31.9 \pm 2.5\%$ increase relative to 4D-; $p < 0.01$). We believe that this is simply explained by the fact that although 4D was not longer expressed after doxycycline, progenitors remained in higher numbers leading to a persistent increase in neurogenesis in the absence of compensatory effects. We observed a similar effect in a parallel study by 4D viral injections in the hippocampus over the life of the animal (Berdugo Vega et al, in preparation).

Referee #3, Minor points:

3) Statistical tests: a) The authors have used multiple individual t-tests to compare groups within a given experiment (e.g. Fig.1C/D, Fig.2E). This approach can generate spurious false positive results in terms of significance. In these cases, and potentially elsewhere in the paper, the authors should perform 2-way ANOVA analyses (which will probably reveal a simple overall effect of 4D+ across all cell types).

Authors response

We have performed ANOVA on the data in Fig. 1 and 2 as requested by the reviewer with results comparable to our previous analysis (see below).

4D- vs 4D+ parameter	new ANOVA	previous t-test
BrdU/EGFR+/Mash1-	F(1, 4)=33,56, p<0.004	p<0.003
BrdU/Nestin+/S100B-	F(1, 4)=31,03, p<0.005	p<0.002
BrdU/Mash1+	F(1, 4)=14,72, p<0.019	p<0.008
BrdU/DCX+	F(1, 4)=10,97, p<0.029	p<0.009

Although it is true that significance decreases, this effect is negligible and bears no consequence on our conclusions. We believe however that when comparing cells in Fig. 1-2, t-test is more appropriate because markers were analyzed on independent stainings comparing only two groups (4D- vs 4D+) with no influence by repetitive measurements of the same samples nor time. Taking this into account, we still consider t-test formally correct.

Referee #3, Minor points:

b) Statistical tests for differences in proportions (e.g. Fisher's Exact Test) should be performed for the data in Fig.2C/E

Authors response

We agree and thank the reviewer for this comment now corrected throughout our study.

Referee #3, Minor points:

c) Everywhere, much more comprehensive reporting is needed of all quantitative comparisons: for every test, please report the mean+SEM(or SD) values for each group, test statistic (e.g. t value), df, and actual p value (not just p<0.05)

Authors response

We thank the reviewer for this point and included a new section on statistics in methods (page 22). We also added *p* values throughout the text and additionally indicated the key parameters used for analyses (mean, variance, N=mice and n=cells, etc) in all figure legends.

Referee #3, Minor points:

- Methodological detail: a) For image analysis, how exactly how were cells determined to be positive for individual markers? Were experimenters always blind to group here? Was co-label evaluated in 3d?

Authors response

As now described in Methods, images were analysed throughout z-stacks and maximal intensity projections obtained with an automated apotome or confocal microscope.

Quantification of cellular effects could not be blind because of the need to identify RFP+ cells. In contrast, olfactory tests were performed in a fully automated olfactometer and analysed by an experimenter blind to the manipulation. In essence, the Dresden team assigned mice of either genotype to two counterbalanced groups that were sent to the Heidelberg team to perform olfactometry tests blind. Only at the end of the behavioral tests and the finalization of the analyses were genotypes of each mouse revealed and subsequently confirmed by PCR and RFP immunolabeling on brain sections. 3D reconstruction was not necessary for the colabeling of markers (e.g. Fig. 1 and 2), while this was performed for branching, spine density and Sholl analyses (Fig. 3) as mentioned in the text.

Referee #3, Minor points:

b) The electrophysiology section lacks crucial information on individual protocols, sampling frequencies, quality control (series resistance, capacitance compensation, bridge balance), and how measures were generated (e.g. how was V_m determined? And R_{in} ?).

We have now significantly extended the section on electrophysiology with all the information requested, including the several new measurements performed, in SI (page 5-7 and new Fig. 3E-H and S3B).

Thank you for submitting your revised manuscript to the EMBO Journal. Your study has now been re-reviewed by referee #2 and 3 and their comments are provided below. Both referees appreciate that the analysis has been strengthened, but still have some concerns regarding the conclusion that "increasing neurogenesis improved the discrimination ability of mice challenged with difficult tasks" as you look at "single" very specific behavioural task.

I have discussed this issue further with the referees. Both agree that the manuscript provides important insight but they also find that you have to clearly state in the text that you find improved discrimination for one single odor pair only and avoid overstatements that this has broader applicability to discrimination ability in general. OK to discuss this, but make sure you clearly state what the findings show.

When you submit your revision would you also take care of the following things:

- You have at the moment supplemental figures, but we no longer have such figures. We refer to them as expanded view figures or appendix. Please see author guidelines <http://emboj.embopress.org/authorguide#expandedview> and please also make sure to fix the callout to these figures in the text
- The movie legend needs to be separated from figure legend and zipped to the movie file.
- The reference list needs to be fixed. For citations with more than 20 authors we use 20 authors et al. You currently list 10 authors before et al.
- In Suppl. Fig 2B there is a funny cut at the very bottom. Please take a look
- I have asked our publisher Wiley to do their pre-publication check on the manuscript. They will send me their comments tomorrow. I will pass on their comments as soon as we receive them
- We include a synopsis of the paper that is visible on the html file (see <http://emboj.embopress.org/>). Could you provide me with a general summary statement and 3-5 bullet points that capture the key findings of the paper?
- It would also be good if you could provide me with a summary figure that I can place in the synopsis. The size should be 550 wide by 400 high (pixels).

That should be all - congratulations on a great paper. Let me know if we need to discuss any more specifics.

 REFEREE REPORTS:

Referee #2:

In the revised version of this manuscript, the authors have addressed some of the problems pointed out in my original review.

The key finding of this study, however, that increased neurogenesis improves the discrimination of similar odors, is based on a single behavioral test with a single odor pair.

I do not think that this provides convincing evidence to support the authors' claim.

The difficulty of the odor discrimination task is defined by the mixture composition, the argument that very difficult to discriminate odor pairs are difficult to find appears far-fetched.

Referee #3:

I am impressed by the extra data, attention to detail and effort involved in the authors' responses to my concerns. They have satisfactorily addressed all of the issues I raised, except just one: the fact that the major conclusion from their paper rests on a single, highly specific behavioural task (my comment 1v in the rebuttal). Here I complete accept the authors' argument that it would be hard to find a more difficult olfactory discrimination task, but my point was actually that it is important to replicate their effect in a different task of similar difficulty. If it were me, I would want to be very sure that the result determining the title of my paper was not just a one-off finding. Nevertheless, I entirely take the authors' point that to generate such extra data would be excessively costly in terms of time, money and animals. As a compromise, I suggest very strongly that the authors change the sentences in their abstract (currently '...increasing neurogenesis improved the discrimination ability of mice challenged with difficult tasks...') and results (currently '...increasing discrimination performance specifically in extremely challenging tasks.') to make it abundantly clear that they only ever observed group differences in olfactory performance on one single task, not on difficult or challenging 'tasks' in general.

2nd Revision - authors' response

14th Nov 2018

Point-by-point response to the revised manuscript EMBOJ-2017-98791R

Referee #2:

In the revised version of this manuscript, the authors have addressed some of the problems pointed out in my original review. The key finding of this study, however, that increased neurogenesis improves the discrimination of similar odors, is based on a single behavioral test with a single odor pair. I do not think that this provides convincing evidence to support the authors' claim. The difficulty of the odor discrimination task is defined by the mixture composition, the argument that very difficult to discriminate odor pairs are difficult to find appears far-fetched.

It is correct that we have used a single difficult task (“difficult” being defined as mice being unable to reach performances of 95%) and agree that referring to this in the text in the plural form as “difficult tasks” was inappropriate. We have therefore changed this throughout the text, as reviewer 3 requested, to make it abundantly clear that this was a single task (see reply to reviewer 3 and red text in the manuscript). We would also like to point out that our statement that “this was among the most difficult odor pairs for mice to discriminate” was not based on speculation but on data described in our previous point-by-point response and cited article (<https://doi.org/10.1101/356279>).

Referee #3:

I am impressed by the extra data, attention to detail and effort involved in the authors' responses to my concerns. They have satisfactorily addressed all of the issues I raised, except just one: the fact that the major conclusion from their paper rests on a single, highly specific behavioural task (my comment 1v in the rebuttal). Here I complete accept the authors' argument that it would be hard to find a more difficult olfactory discrimination task, but my point was actually that it is important to replicate their effect in a different task of similar difficulty. If it were me, I would want to be very sure that the result determining the title of my paper was not just a one-off finding. Nevertheless, I entirely take the authors' point that to generate such extra data would be excessively costly in terms of time, money and animals. As a compromise, I suggest very strongly that the authors change the sentences in their abstract (currently '...increasing neurogenesis improved the discrimination ability of mice challenged with difficult tasks...') and results (currently '...increasing discrimination performance specifically in extremely challenging tasks.') to make it abundantly clear that they only ever observed group differences in olfactory performance on one single task, not on difficult or challenging 'tasks' in general.

We are very pleased by the words of this reviewer who appreciated our new data and efforts in addressing all remaining concerns. We do agree, and the reviewer is entirely correct, that the difficult task being investigated was just one and hence we have changed all statements throughout the text to reflect this including 2 sentences in the abstract, the last sentence of results and the discussion. These changes are visible in red text.

Corresponding Author Name: Federico Calegari

Journal Submitted to: EMBO J

Manuscript Number: EMBOJ-2017-98791R